# Timely deployment of best-in-class technologies to enable development and decarbonise construction

Cyrille Dunant [1], Hisham Hafez [2], Alastair T. M. Marsh [3], Sabbie A. Miller [4], Martin Röck [5], Wolfram Schmidt[6], Karen L. Scrivener [3] ✉ & Franco Zunino [7]

In the face of two apparently irreconcilable global challenges - housing a growing world population and reducing $CO_2$ emissions - we analyse the current, historic and forecast data on the use of construction materials. Today, cement-based materials make up around three quarters of materials used by mass. Historically, we see that cement-based materials use goes through a peak as Gross Domestic Product per capita increases and then falls. This peak of cement use has been particularly pronounced in China, but is now on a downwards path. From now to 2050, three quarters of construction materials demand will be in low- and middle-income countries. We estimate that adopting the best available construction technologies could reduce $CO_2$ emissions by about 73% compared to business as usual by 2050. In low- and middle-income countries, the housing and infrastructure needed to achieve the Sustainable Development Goals could be supplied while simultaneously reducing their per capita $CO_2$ emissions from structural materials.

Climate change is among the most urgent challenges of the next decades, with recent findings suggesting we are at critical levels of warming[1]. To meet the Paris Agreement target of limiting the average temperature increase to ≤1.5 °C, we need to surpass existing commitments and approximately halve $CO_2$ emissions by 2030 and achieve net zero $CO_2$ emissions by 2050[2]. Decarbonisation of construction materials has received a notable amount of attention from both academic authors[3,4] and industrial organisations[5,6], with a particular focus on cement & concrete and steel. Yet the specific question of what degree of decarbonisation is achievable at the global level, via readily implementable strategies, is less well-addressed. Zhong et al.[7]. modelled material decarbonisation achievable in the buildings sector (residential and commercial) up to 2060 for seven construction materials, using a range of seven material efficiency strategies; a reduction of ~62% was deemed achievable by 2050 (from a 2020 baseline of 3.17 gigatonnes of $CO_2$ per year (Gt.$CO_2$/year) to 1.21 Gt.$CO_2$/year in 2050). Pauliuk et al.[8]. focussed on residential

buildings, and they studied the interactions of applying four materials efficiency strategies with different socio-economic pathways and background climate policies; for the reference policy and socio-economic pathway (i.e., SSP2), a reduction of ~49% was deemed achievable by 2050 (from 8.7 Gt.$CO_{2(eq.)}$/year in 2020 to 4.4 Gt.$CO_{2(eq.)}$/year in 2050). The larger absolute emissions values in this study are due to the inclusion of operational emissions in additional to embodied emissions. The Global Resource Outlook 2024[9] took a holistic approach to decarbonisation pathways integrating material efficiency, energy policy and societal trends; whilst headline findings are clear, the broad scope makes it difficult to isolate the specific assumptions and mitigation contributions in construction materials production.

These previous global studies give an encouraging picture for decarbonisation, but they have two key limitations. First, studies used frameworks that account for material demand from buildings (i.e., Resource Efficiency Climate Change mitigation framework (RECC)[10] in

[1]Engineering Department, University of Cambridge, Cambridge, UK. [2]School of Civil Engineering, University of Leeds, Leeds, UK. [3]Laboratory of Construction Materials, Ecole Polytechnique Federale de Lausanne, Lausanne, Switzerland. [4]Civil and Environmental Engineering, University of California, Davis, CA, USA. [5]RISE Institute for Regenerative Spatial Systems Science, Vienna, Austria. [6]Bundesanstalt für Materialforschung und -prüfung, Berlin, Germany. [7]Department of Civil and Environmental Engineering, University of California, Berkeley, CA, USA. ✉e-mail: karen.scrivener@epfl.ch

Pauliuk et al.[8] Integrated Model to Assess the Global Environment (IMAGE)[11] in Zhong et al.[7].), but not from infrastructure and other uses of the materials. Projecting material demands from infrastructure is more challenging, yet it is crucial for obtaining an accurate global picture. A significant global proportion of construction materials is already consumed in infrastructure (approximately 40% for construction steel[12], 33% for cement[13]), and large investments in new infrastructure are seen as essential to sustainable development in low and middle income countries[14]. Second, some strategies evaluated are highly uncertain within the 25-year timeframe to 2050. For example, both Zhong et al.[7]. and Pauliuk et al.[8]. included reduction in per person floor area as an emissions reduction strategy; this is a potentially powerful material efficiency lever, but its implementation is dependent on highly uncertain social, policy and market factors[15].

Two ongoing (yet unresolved) debates related to decarbonisation are the extent to which absolute decoupling is possible[16,17], and how pollution and consumption evolve with increasing national income (i.e. the disputed existence of the Environmental Kuznets Curve[18,19]). These debates are critical to questions around climate equity (i.e. how much should each country be allowed to emit?), and the synergies and trade-offs between decarbonisation and development (i.e. how should decarbonisation strategies differ for lower and middle income countries compared to high income countries?). Whilst the previous studies around material decarbonisation strategies do assess differences in stock and flow characteristics between different world regions and/or income brackets, they have not addressed how consumption trends relate to Gross Domestic Product (GDP) per capita at the granularity of national level data.

At the same time as the need to decarbonise materials production, there is a pressing need for more construction to accommodate the housing and infrastructure needs of a growing and urbanising world population, particularly in lower and middle income countries. For example, Sustainable Development Goal (SDG) Target SDG 11.1 is "*By 2030, ensure access for all to adequate, safe and affordable housing and basic services and upgrade slums*" and adequate housing is a prerequisite for many other SDGs. Yet in 2023 it was estimated that over 1.6 billion people still lived in inadequate housing[20]. It has been estimated that the equivalent of New York City needs to be built every month[21], and >50% of global floor area additions (corresponding to ~125 billion m²) from 2017-2050 are projected to take place in lower and middle income countries across Africa, India and China[22].

A vast array of technological solutions has been proposed to decarbonise construction materials. However, many of these solutions do not meet realistic constraints to be affordable, scalable and adaptable within construction sector supply chains on a timescale needed to meet climate goals. Solutions must meet both the challenges of decarbonisation, and society's urgent housing and infrastructure needs at scale – tackling only one of these challenges is not sufficient. Given the conservative culture and slow-moving nature of the construction sector, a key question is how much of the conventional construction materials are likely to be used from now until 2050, and what level of decarbonisation could be achieved by applying feasible, yet not widely deployed, technologies.

In this study, we aim to give a comprehensive answer to the question of what degree of decarbonisation in construction materials is possible using readily implementable strategies, already deployable within the materials and construction value chain. And, to contribute fresh insights to long-running debates around decoupling and development. To achieve this, we analyse the potential evolution of the use of construction materials to 2050 from two complementary perspectives (using the top-down/bottom-up terminology by Lanau et al.[23].): first, from analysis of the historic trends of use of the major structural materials as a function of national economic progress expressed as GDP (i.e. a top-down approach); then, from the perspective of the projected

evolution in floor area of buildings, globally (i.e. a bottom-up approach). We focus on material flows in this study, rather than stocks, as material flows are directly relevant to embodied carbon emissions of newly produced materials. In terms of materials, we focus on cement-based materials, steel and timber as these make up the majority by mass of structural materials used in construction.

## Results

### Current production of construction materials

In Fig. 1a, we present an analysis of the major materials produced in 2019 (details and assumptions presented in the Methods and Supplementary Methods) by mass, volume, and $CO_2$ emissions. The dominance of construction materials is clear (Fig. 1b), accounting for ~33 Gt out of a total material production of ~37 Gt. 'Cement-based materials' includes all Portland cement-based products, including concrete, mortar, and prefabricated concrete products including blocks, pavers and tiles. Cement-based materials – almost exclusively used in construction – alone make up around three quarters of all materials produced by mass. Bricks and asphalt are also used exclusively in construction along with ~55% of steel, with lower proportions for glass ( ~ 39%), aluminium ( ~ 24%), plastics ( ~ 20%) and wood ( ~ 11%). While construction materials contribute ~90% by mass of all materials production, their share of embodied $CO_2$ emissions from all materials is much lower ( ~ 56%). Nonetheless, globally, the production of construction materials contributes approximately 14-17% of annual anthropogenic $CO_2$ emissions (6.4 Gt.$CO_2$, as calculated here, out of a reported $45 \pm 5.5$ Gt.$CO_2$ in 2019[24]). Notably cement-based materials account for around 5.5-7.5% of all anthropogenic $CO_2$ emissions.

Structural materials (i.e. cement-based materials, steel, brick and timber) make up ~93% of all construction materials by mass; in comparison, materials mostly used in buildings for the envelope (i.e. glass, plastics, aluminium) make up only ~0.5% by mass. Similarly, structural materials account for the majority of $CO_2$ emissions from construction materials ( ~ 87%), compared to only ~11% from envelope materials. Whilst there is a need to decarbonise building envelope materials, the net potential for decarbonisation is far greater for structural materials.

### Top-down approach based on historic trends of consumption

We analysed the evolution per capita of the main structural construction materials (cement-based materials, steel, timber) for all countries as a function of GDP per capita (Fig. 2) (with a few exceptions, as detailed in the Methods section). While virtually all cement-based materials are used in construction, this sector accounts for only about half of the total steel[12]. For timber, an estimated 11% of harvested timber is used in construction (see calculations in Supplementary Methods). The data underlines the fact that cement-based materials are by far and away the most widely used type of material.

It is seen that all countries initially show an increase in materials use per capita with GDP/capita. However, cement-based materials consumption peaks at around $20,000/capita (US Dollars) and then decreases. Steel is more variable, likely reflecting differences between countries with large industrial bases and those without, but seems to peak at around $30,000/cap. These trends are consistent with those identified by Bleischwitz et al.[25], on domestic consumption of cement and steel for Germany, Japan, United Kingdom, United States of America and China. Although the pattern of development is very predictable, the heights of the peaks vary considerably. Curves for cement and steel for selected individual countries are presented in Supplementary Figs. 1 and 2. Timber shows the most variation, with no discernible peak in the data. Unsurprisingly, major timber producing countries, such as Canada and the Nordic countries show much higher use, whereas e.g., the Middle East has only a modest consumption. Most striking is the very high peak of cement consumption in China,

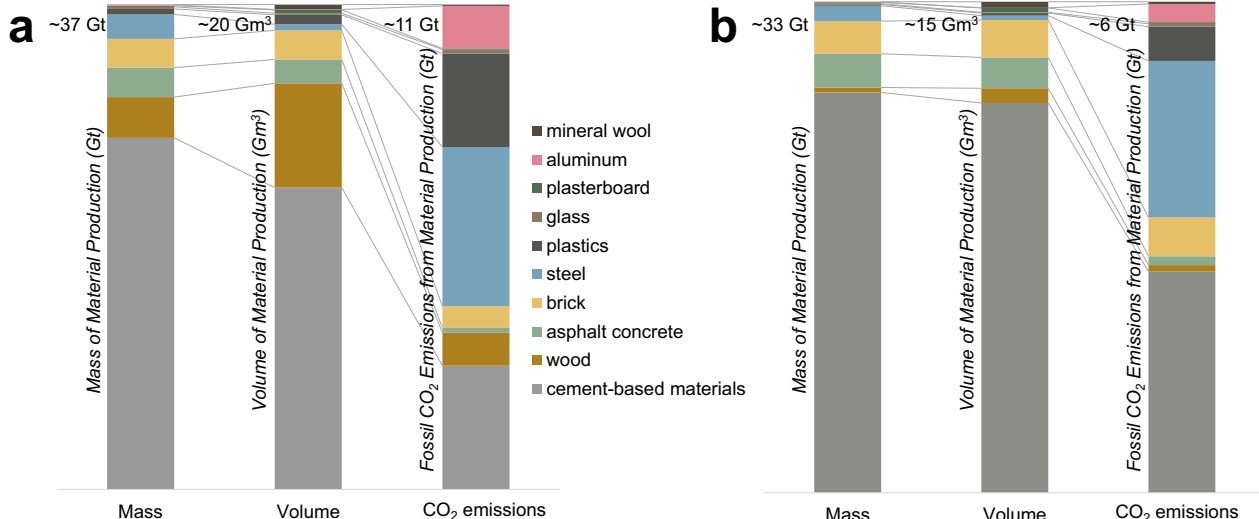

**Fig. 1 | Approximate production of materials in 2019 and their associated CO₂ emissions.** Data is presented separately for **a** all production, and **b** use in construction. Data are presented in terms of mass produced (Gt), volume produced (billion m³), and $CO_2$ emissions (Gt) from production. Use phase $CO_2$ emissions or uptake (i.e., fluxes) and end-of-life fluxes are not represented, Scope 2 emissions occurring from production are included for steel, aluminium and cement, but Scope 3 $CO_2$ emissions are excluded. Additionally, carbon-uptake by biomass during cultivation is not reflected. "Wood" products include all roundwood; asphalt concrete includes bitumen binder and aggregates; "Cement-based materials" includes all Portland cement-based products (i.e., including concrete, mortar, cement-based blocks, pavers, and tiles, etc.). Data sources and modelling assumptions for the data presented in this figure are presented in the Methods section, and numerical values for the charts are provided in Supplementary Table 1.

although the peak was already reached in 2014 at a GDP/capita of around $10,000.

Previous works have sought to establish a point at which material saturation is reached, with countries having built a sufficient stock[25–27]. However, these studies show considerable variation in the level of stock at which saturation is reached, and more recent analysis has cast doubt on whether stock saturation is truly observed[28]. Our data suggests rather that the peak material use reflects more a level of wealth or development for a given country than some level of material saturation per se. The historical trends in Fig. 2 show a relative decoupling of GDP and material use beyond a certain GDP per capita level, consistent with a general picture of relative decoupling conditions built up over numerous country-level studies[17]. Regardless of the point at which stocks saturate[28], the inflows of newly produced materials are the more critical issue in terms of decarbonisation. The 'Environmental Kuznets Curve' (EKC) postulates that the scale of environmental impacts follow an inverse U function with regards to GDP per capita[29]. This concept has since been investigated with regards to materials consumption, with studies presenting evidence both in favour of[30,31] and against[32] the existence of a 'Materials Kuznets Curve'. The profile of the global master curves for steel and cement (Supplementary Fig. 3) agrees well with the profile of the 'conventional EKC', in which increasing income leads to a decrease in impacts after a peak (but not a full decline)[19]. This global trend, consistent with the EKC concept, also aligns with previous findings: that in low- and medium-income countries, the capital formation is largely associated with urbanisation and building of infrastructure; whereas in medium-income and advanced economies, it takes the form of dematerialised investments and tooling[33].

Projecting GDP/capita and population to 2050 suggests that the world will then be just past its cement peak, and a few years behind its steel use peak. By mid-century, on current trends, the cement consumption will be 5 Gt (25–50 Gt of cement-based materials), 3.8 Gt of steel (1.5 Gt in construction) and 16.2 billion m³ (or 11.83 Gt) of roundwood (1.29 Gt in construction). This consumption of cement-based materials corresponds to a worldwide trend lying between the

development histories of US and Germany. If the world were to follow a path similar to China or South Korea, the cement consumption in 2050 would be closer to 15 Gt.

## Bottom-up approach based on projections of added floor area

To test the robustness of the 'top-down' estimates, a complementary 'bottom up' estimate was carried out. Material consumption in buildings, including structural, non-structural and envelope functions (cement-based materials, steel, timber, bricks), was estimated from projections of the growth in global floor area (m² floor area / year) to 2050 from Deetman et al.[34], in combination with representative material intensity values (kg/m² floor area) from a variety of other data sources (assumptions presented in the Methods section). Quantities for infrastructure materials (cement-based materials, steel, mortar, asphalt) were estimated pro-rata, using known proportions for the global use of each material in buildings and infrastructure (see Methods).

The mass distribution of construction material consumption in the reference year 2019 and the projection in 2050 mainly consists of cement-based materials (21.7 Gt, 29.4 Gt), brick (2.6 Gt, 3.5 Gt) and asphalt (1.95 Gt). Steel (0.82 Gt) and timber (0.81 Gt) make up minor quantities. The scale of increase ( ~ 43%) in consumption of cement-based materials to 2050 is similar to that projected by the Global Cement and Concrete Association[6], and not far from the ~60% increase of construction material inflows into construction in the Global Resources Outlook 2024[9]. Figure 3a summarises the 'top-down' and 'bottom-up' results for cement-based materials, steel and timber). The similarity of the values from the two approaches indicates that, while it is impossible to calculate precise future consumption, the estimated trend is robust.

From the 'bottom-up' dataset it was possible to predict the demand for cement-based materials across different regions (details in the Methods section). The highest demand is projected to occur in regions with average GDP per capita in the range of $10–20k (Fig. 3b), with about 75% in lower- and middle-income countries (according to World Bank definition of high income as above $14,000 GDP/capita) (Fig. 3c). These data reinforce the critical role of

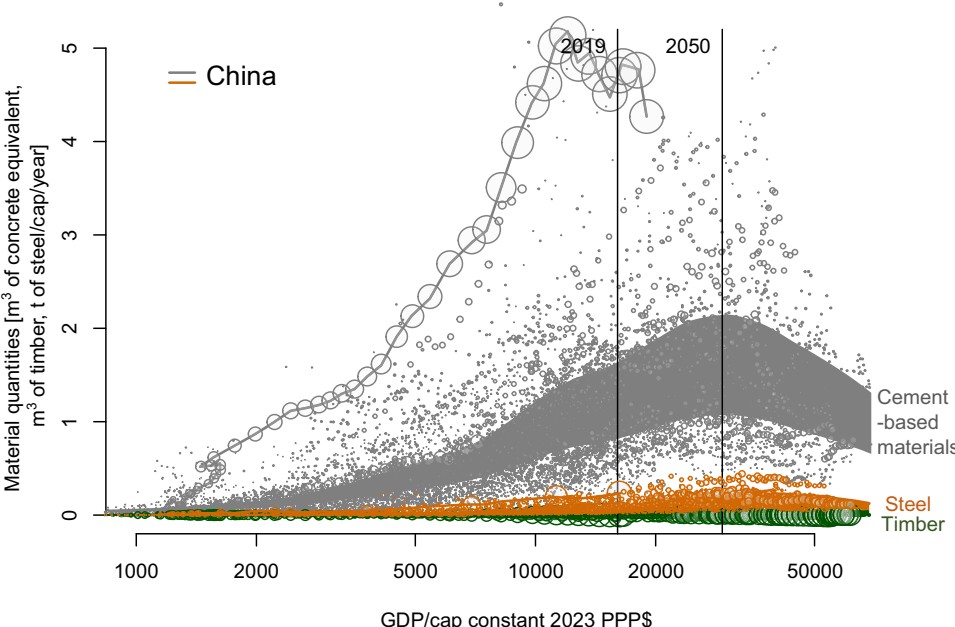

**Fig. 2 | Historical data for the consumption of the main materials used in construction (cement-based materials, steel and timber).** The bands indicate the uncertainty ranges, as described in detail in the Methods section. For cement-based materials, the uncertainty represents the range of cement content in cement-based materials from 200 to 500 kg/m³. For steel, the uncertainty represents the variable share of steel consumption in construction worldwide from 25 to 55% (of total steel consumption). For timber, the uncertainty represents the variable share of timber consumption in construction worldwide from 2.6 to 15.1%.

lower and middle income countries in decarbonisation of cement-based materials[35].

## Potential of mitigation approaches to lower CO₂ emissions

In this section we look at the potential of technologies which are technically feasible and readily implementable today at a global scale, to lower the CO₂ emissions from cement-based materials and steel in 2050. We excluded material efficiency strategies which we deemed to be either too unconventional for widespread adoption (e.g., novel structural systems) or to depend on external market, policy and social factors (e.g., reduction in per person floor area). These calculations are based on the projected quantities established from the top-down approach above. The approaches studied are summarised in Fig. 4. A brief overview follows, with a more detailed description of rationale and input values given in the Methods and Supplementary Methods. There are many uncertainties in this analysis as discussed briefly here and more in the Methods section, but we have tended towards conservative estimates, so higher savings could be possible in many cases.

### Substitution with timber and other bio-based materials

Substitution range is assumed to 4–6% for cement and 2–3% for steel, reflecting maximum and minimum production scenarios from the Food and Agriculture Organisation of the United Nations (FAO)). The numbers reflect the mass of a timber section replacing a steel section of equivalent load-bearing capacity, and in the case of concrete, assume that in general 1 m³ of timber can substitute 1 m³ of concrete, assuming 2180 kg/m³ as the specific mass of concrete and 730 kg/m³ for the specific mass of timber. The embodied carbon for timber used (0.25 t.CO₂/t) is a well-established value from the Institution of Structural Engineers[36], yet represents a favourable accounting approach as it excludes the embodied carbon associated with harvesting (see Supplementary Methods for further explanation). The extent to which structural bio-based materials could displace concrete is limited by the quantities of bio-based materials that can be harvested sustainably. We estimate that the CO₂ reduction from increasing timber production to replace part of concrete is very small: at the current rate of growth of

timber production, no more than 5% of the projected use of concrete could be substituted, considering the current difference in size of the timber and concrete markets (Fig. 1) coupled with the projected growth of the concrete market (see Supplementary Methods for more details).

### Building structural design

Mitigation range is assumed to be 13–43% lower steel and cement emissions (detailed explanations provided in the Supplementary methods). Many studies have shown that considerable reductions in material quantities (for the same floor area) are possible at the design stage for built systems – this lowers the material intensity coefficient (i.e. the mass of material per unit of floor area) of the structure, and hence lowers the embodied carbon of the structure. The choice of frame type, layout, and decking is commonly very suboptimal due to the wide range of possible choices and their comparatively low price sensitivity. Dunant et al. found that CO₂ emissions could be reduced by as much as 50% by better early stage design choices, but a central estimate factor of 25% has been assumed to allow trade-offs for the sake of function, prestige and aesthetics[37]. Such values have been used previously as global[6] or regional averages[38].

The factors have been applied uniformly worldwide, because they are relative factors: design practice is inefficient because the task of choosing the appropriate structural form for purposefully lean design is difficult[37]. As countries urbanise, the architectural forms will tend to become more material efficient due to the adoption of mid-rise typologies (i.e., ≥3 floors) over low-rise (i.e., 1–2 floors), independent of the quality of the design. They will, however, become more concrete and steel intensive. The drive towards more professional design and the universal availability of design tools justify the average 25% estimate for a reduction in material use from the baseline.

### More efficient concrete production

Mitigation range is assumed to be 15–40% of cement emissions. As shown by Damineli et al.[39] and later by Zunino[40], there is a large scatter in the amount of binder used–and thus embodied CO₂–for the same

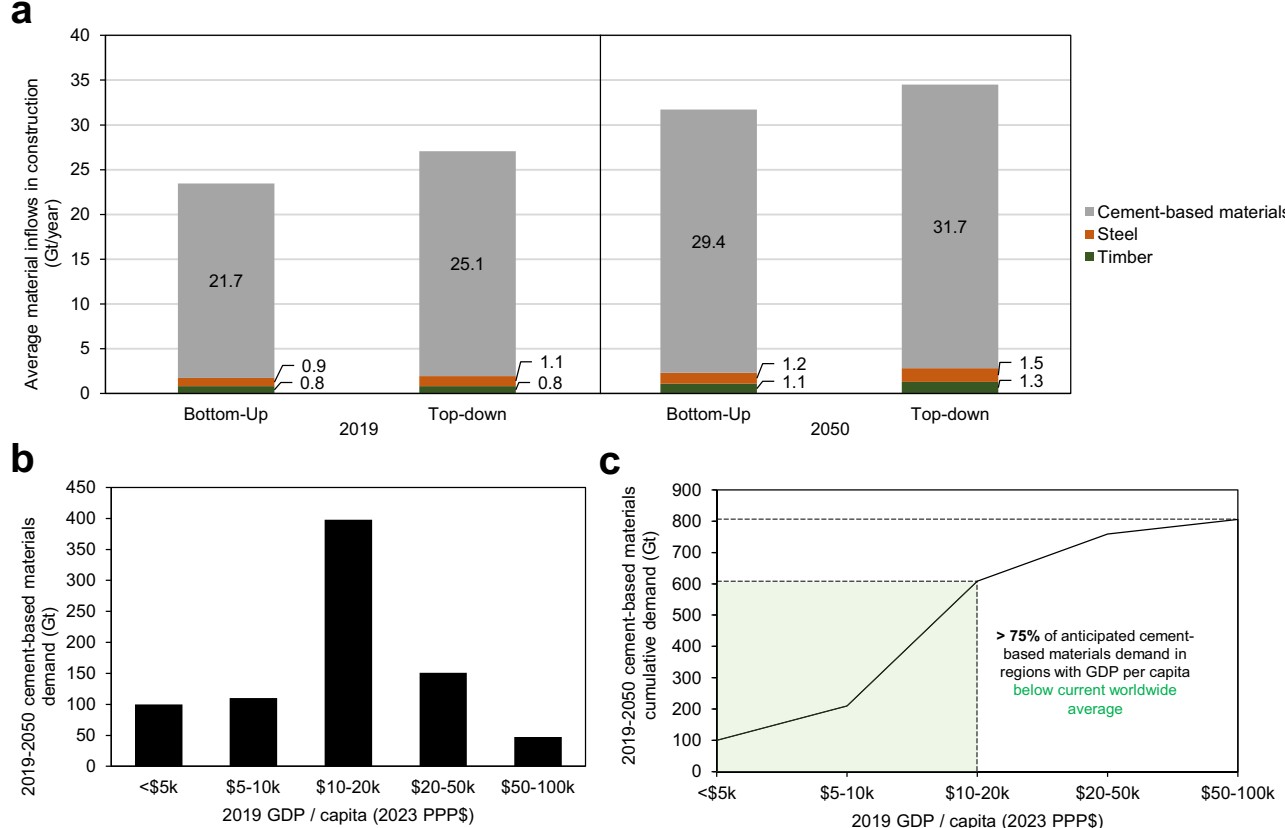

**Fig. 3 | Projections of future construction material demand and its distribution between country-based income groups. a** Comparison of estimates by top-down and bottom-up approaches to project annual consumption of structural construction materials in 2050. **b** Distribution of projected 2019-2050 cement-based materials demand by wealth intervals of different regions. **c** Cumulative distribution of projected 2019–2050 cement-based materials demand by wealth intervals of different regions. $ in x-axes of b and c refers to US Dollars.

strength class. A large fraction of the variation can be attributed to the informal nature of concrete production. In general, moves to formalised processes and professional operations will lead to lower $CO_2$ emissions of concrete. There is no systematic correlation between durability performance and binder content[40], highlighting the need for updating standards that anachronistically prescribe a minimum cement content based on exposure conditions. By reducing from typical cement dosage of 350 kg/m³ to a conservative 280 kg/m³ (within EN-206 standard limits), the carbon footprint can be reduced by about 20%. Used in combination with higher amounts of supplementary cementitious materials in cement (described below), better concrete production could yield an overall $CO_2$ reduction of at least 45% (more details in Supplementary Methods).

**Extending use of supplementary cementitious materials**
Mitigation range is assumed to be 15–40% of cement emissions. Lowering the content of clinker in cement is a very successful strategy to lower the embodied $CO_2$ emissions of concrete. Currently, the average level of clinker substitution is only around 29% worldwide (2018 estimate)[41]. A major limit to the level of substitution has been the lack of availability of the two most commonly used supplementary cementitious materials (SCMs): blast furnace slag and suitable fly ash from coal powered electricity generation. Today, global supplies of blast furnace slag and suitable fly ash are limited to around 17% of cement production[42], and this value is likely to decrease as we move away from the use of coal and as the steel industry also decarbonises. However, new sources of substitute material are now becoming available using calcined clays and "Limestone Calcined Clay Cement" (LC³) technology[43]; kaolinitic clays suitable for calcined clay

production can be found within all world regions[44]. With these new SCMs, an average substitution level of 40% by 2030 would be possible and maybe a level above 50% by 2050, with research to solve some of the issues which low clinker contents present (for example, low early strength and placeability).

**Steel decarbonisation**
Mitigation range is assumed to be 42.5–90% of steel emissions. Steel can be electrically recycled in electric arc furnaces (EAF), and with grid emissions reducing, a likely practical limit of 100 kgCO₂/tonne can be achieved with the residual emissions linked to carbon injections, electrode consumption and lime use (a detailed explanation is provided in the Supplementary Methods). This value is much lower than the current 1900 kg $CO_2$/tonne which is the average (inc. Scope 1 and 2) emissions of steel in the world[45]. Recycled steel cannot be made in all grades, and it is frequently used for producing grades with the lowest strength requirements[46]. According to the IEA, 47–57% of steel will come from EAFs or direct reduced iron (DRI)-EAFs in 2050[5]. Construction can use any type of steel and will be able to absorb all of the recycled steel produced[47]. Therefore, a practical factor of 65% abatement has been applied.

**Circular use of concrete fines**
Mitigation range is assumed 2.5–15% of cement emissions. Most concrete recycling is the use of crushed concrete from demolition as replacement aggregates. This practice does not abate $CO_2$ in general (and frequently raises it, because low quality aggregates require more cement). However, cement can be separated from concrete at the end of life[48] and the calcium-rich fines used as raw meal for cement, saving

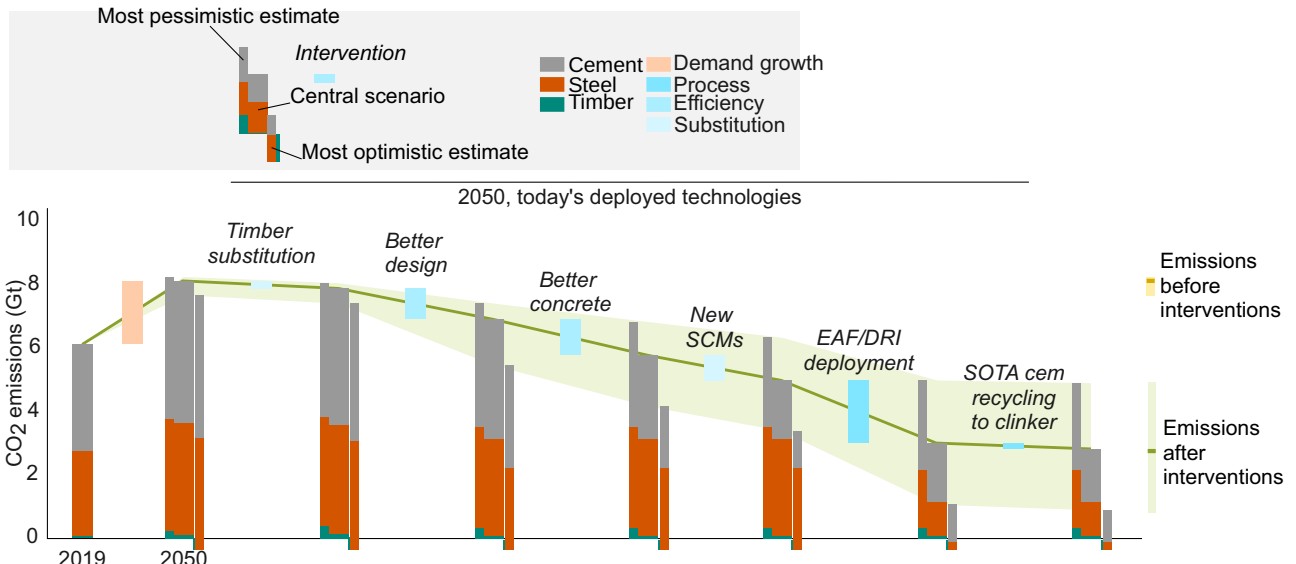

**Fig. 4 | The decarbonisation potential for cement-based materials and steel production in 2050, based on technologies which are feasible today.** EAF electric arc furnace, DRI direct reduced iron, SOTA state of the art, SCM, supplementary cementitious material. The error band represents the range between the most pessimistic estimates and the most optimistic estimates.

fuel and process emissions[49,50] but limited to perhaps 10% emissions abatement. In the future, higher savings could be possible using electric kilns[51] (see Supplementary Methods for further explanation).

## Discussion

The analysis in this paper clearly indicates that cement-based materials will continue to be the dominant materials for construction by mid century. About three quarters of demand will be in low or middle income countries, where construction materials are urgently needed to build housing and infrastructure for the growing population and meet the Sustainable Development Goals. The capacity of timber to replace concrete and steel is severely limited by the amount which can be harvested sustainably. On the other hand, the majority, around 70%, of $CO_2$ emissions from steel and cement-based materials production in 2050, could be abated by using currently accessible solutions (Fig. 4). This is a very similar value calculated in previous studies[52]. These feasible and proven approaches yield a carbon saving potential that is roughly 60-to-80 fold the substitution benefit estimated from increased use of wood for construction (see Supplementary Methods), even when favourable values for the embodied carbon of timber are used which do not account for the carbon emissions arising from harvesting.

Compared to a business-as-usual scenario in 2050, a 73% saving is achievable; however, taking into account the increase in demand between 2019 and 2050, the reduction in $CO_2$ emissions in comparison to the reference year of 2019 is ~55%. This value lies within the range of reductions (~45-87%) recommended for 'transport, industry and buildings' in the 1.5 °C pathway from the Intergovernmental Panel on Climate Change (i.e. a reduction of compared to 2015 levels)[53]. Whilst the ultimate goal remains Net Zero for construction materials, further decarbonisation will require more costly measures (e.g., CCUS) or more uncertain measures that depend on social factors (e.g. floor area reductions).

In a global evaluation considering all sources of greenhouse gas emissions, Global Resource Outlook 2024 projected that by 2060 (relative to a reference year of 2020), the reductions in emissions due to current technology trends (-51.0 Gt.$CO_{2(eq.)}$/year) would largely be cancelled out by the increase in emissions due to increasing affluence and population growth (+62.8 Gt.$CO_{2(eq.)}$/year)[9]. As a consequence, net reductions (with regard to the 2020 reference year) were only

deemed achievable through policy and societal shifts. In contrast, our study indicates that within the sector of construction materials a much higher degree of relative decoupling is achievable within current feasible technologies, beyond which strategies relying on major policy shifts and societal changes are needed.

Figure 4 also identifies there is still a gap of around 30% of emissions for which other solutions are needed to reach net zero. Most current scenarios imagine this to come from Carbon Capture, Utilisation and Storage (CCUS), electric kilns or use of hydrogen produced from electrolysing water using excess electricity production from renewable sources[6,54]. However all these solutions have a very high cost (e.g. CCUS is estimated to cost in the range of $60–130 per ton of clinker[34] increasing its cost 2-4 times), and also entail considerable technological, implementation and societal challenges[4].

These challenges also raise the importance of further work on long-term strategies considering socio-behavioural dimensions for enabling reduced material demand while ensuring decent living standards[55,56]. For example, extending the lifespan of buildings to reach closer to their technical lifespan; this will require tackling social and economic incentives for premature demolition, especially for high-income countries with broadly stable building stocks. Most importantly, the readily available strategies for reducing concrete emissions, which are also fairly low cost, should be implemented as a priority. Although reaching net zero emissions in construction is not possible using the technologies which are available today, the potential for reduction is still much larger than commonly understood, leaving only a small gap to be filled using breakthrough or experimental technologies.

In Fig. 5 we look at the implication of these findings for the evolution of emissions from cement-based materials and steel in individual countries as a function of their signalled development path (details provided in Methods). Figure 5a shows the scenario for business as usual (BAU) use of cement-based materials. Most low- and middle-income countries will show an increase in $CO_2$ emissions due to cement-based materials use as indicated by the red lines, even though high-income countries will see decreases in emission due to reduced demand (green lines). Most notably China is likely to show a threefold reduction in cement-based materials related $CO_2$ emissions even with continued growth in GDP. The optimal deployment of just three available technologies: better design, lowering clinker in cement

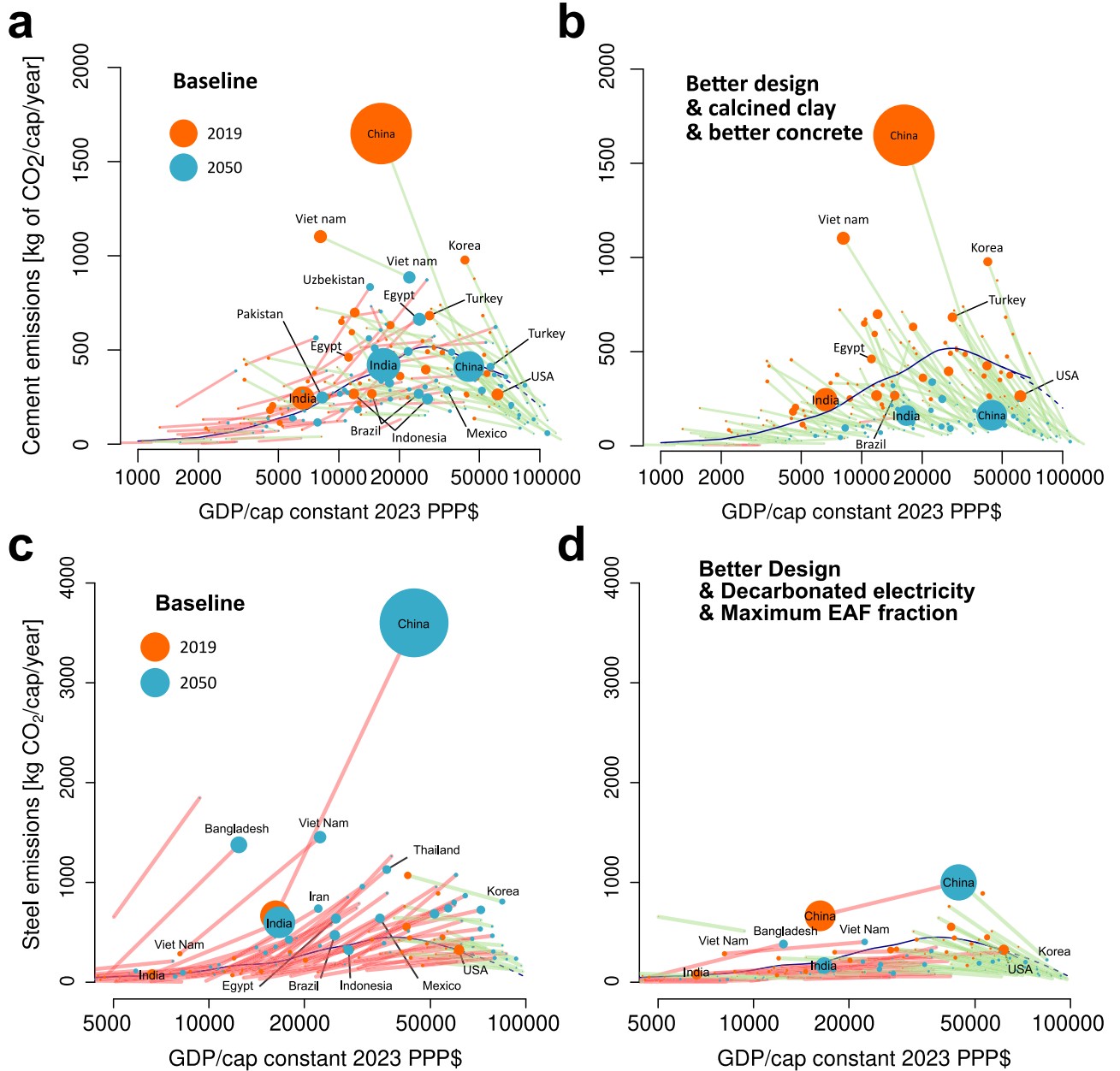

**Fig. 5 | Country by country extrapolation of baseline projections under current trends from 2019 to 2050.** Data is presented separately for: **a** cement consumption emissions per capita, and **c** steel consumption emissions per capita. The trends if all mitigation strategies considered in Fig. 4 are applied are then presented for: **b** cement and **d** steel. Green lines mark decreased emissions, whereas red lines mark increased emissions. The black line on each plot represents the master curves for cement or steel, as shown in Supplementary Fig. 3. $ in x-axes refers to US Dollars.

through use of calcined clay and more efficient production of concrete could dramatically change this picture (Fig. 5b). Even if low- and middle-income countries continue to develop as planned the overall cement-based materials related $CO_2$ emissions can be reduced, with green lines for nearly all countries apart from India where a slight increase may still occur.

For steel production, the situation is not quite as dramatic, but even here the large increases predicted in a BAU scenario (Fig. 5c) could be greatly reduced by maximal deployment of EAF technology plus decarbonisation of the electricity supply (Fig. 5d).

In high income countries, more material-efficient building design and engineering can save up to 50% of materials[37] – these practices are feasible, but need greater incentives to achieve widespread adoption by practitioners. The same is true for better concrete formulation. In lower and middle income countries, the picture is less certain because

much construction is in the informal or self-built sectors, whereas concrete efficiency strategies have mostly been developed for the formal sector with ready-mix concrete. The informal construction sector is globally significant in cement-based construction, especially for low-income housing. As of 2020, an estimated 1 billion people live in informal settlements worldwide, with 56% of the urban population of Sub-Saharan Africa living in informal settlements[57]. There needs to be greater focus on how low-carbon innovations can be developed by, and for, the informal sector. A particular focus should be put on strategies implemented at early stages of the concrete supply chain, such as lowering the clinker factor in cement production and the production of ready to use materials such as concrete blocks. Substituting clinker by SCMs has huge potential and applies to all parts of the world and in all construction types. In Africa, where the additional demand for cement and concrete production capacity will be high in the

coming years, developing regional SCM production capacity will help to reduce imports and create local employment[38]. Given that the solutions described are all already practically feasible, there needs to be fast policy changes to make these existing solutions applicable in real life around the world.

Most importantly this analysis illustrates that growing material demands and $CO_2$ emissions can be decoupled to large extent in the field of construction with the implementation of technologies available today. Given this, it makes sense to invest effort in the implementation of concrete decarbonisation strategies, which are distinct for different regions and contexts.

## Methods

### Boundaries for embodied carbon estimates

For estimates and reports of embodied carbon, this work focuses on fossil $CO_2$ emissions. Other greenhouse gases (GHGs) and biogenic carbon emissions or uptake (i.e., fluxes) are outside the scope of this assessment. We focus on fossil $CO_2$ emissions because for the materials sectors evaluated in this study, $CO_2$ is the most significant of the GHGs emitted by far, arising from fuel and process emissions. $CO_2$ emissions (rather than $CO_{2(eq.)}$ emissions) have been used wherever available. Biogenic carbon is considered outside the scope of this analysis. Due to the temporal factors associated with biogenic carbon uptake, residence times in the built environment, and varied effects of end-of-life management strategies, the role of biogenic carbon in $CO_2$ fluxes can be highly varied[58]. Additionally, factors such as the role of land-use changes and proper forestry management can impact carbon benefits[59]. To reduce uncertainty associated with biogenic carbon and because of the pressing need to mitigate fossil $CO_2$ emissions, we focus on these latter emissions. When it comes to the comparison of decarbonisation strategies, these assumptions are deemed to be unrealistically favourable towards timber. Where possible, we include both Scope 1 and Scope 2 emissions for material production. We have taken 2019 as the reference starting year for this study as the year for which data on most materials exists and which was not affected by the COVID 19 pandemic. Detailed modelling assumptions and data sources for each material are outlined in the Supplementary Methods.

### Estimate of global materials consumption and $CO_2$ emissions

Figure 1 presents estimates for the annual production quantities of major material categories, in both mass and volume, and the embodied $CO_2$ emissions associated with their production. The general approach was first to obtain estimates for annual production of each material category by either mass or volume and use reported densities to convert to determine the other. This approach is straightforward for material categories for which there is relatively little variation in physical density between products or sub-categories (i.e. steel, aluminium, glass). For sectors where there can be significant variation in the physical density of different products (i.e. cement-based materials, plastics, asphalt concrete, wood, mineral wool), further assumptions are detailed in the Supplementary Methods. A summary of values is given in Supplementary Table 1. For embodied $CO_2$ emissions, the general approach was to use previously reported International Energy Agency values for entire sectors (e.g., steel) when available. For some material categories, these data were not available, and instead it was necessary to generate a representative carbon factor (i.e., kg.$CO_2$ per kg.material) and multiply this by the annual production mass. Specific calculations and assumptions for embodied $CO_2$ emissions of each material are described in the Supplementary Methods.

### Top-down estimation approach

The data presented in Fig. 2 were collated from historical data sources for consumption of cement, steel and timber, and GDP per capita. The data used are from the database of cement production from The

Global Carbon Project CEMent-process emissions dataset (GCP-CEM)[41]. This database does not report cement production directly, but it uses a constant emissions factor, so production was determined by dividing by this factor. This emission factor was back-calculated by comparing the US cement-related emissions with the production as reported by the United States Geological Survey. This both validated the fact the emisssions factor was constant and gave the emissions factor. The GDP and population data were from the data tables from Gapminder.org[60], and the projections to 2050 are from the same source. The range of consumption of cement-based materials (as plotted in Fig. 2) was generated by applying a range of cement content from 200 to 500 kg per m³, based on the range of values for cement-based materials presented in Supplementary Table 2.

The data for apparent steel use on a country-by-country basis is taken from the Worldsteel Statistical yearbooks from 1968 onwards. This covers all steel use and is not specific to construction steel. The specific trend for construction steel would be expected to broadly follow the cement one. To estimate the proportion of steel production used in construction (as plotted in Fig. 2) we use a range of 55–25%. The 55% value corresponds to a global average estimate (Fig. 1 of [12]), and the 25% value corresponds to the value for the UK as representative of a high income country (Supplementary Fig. 2 of [61]). This range of values also agrees with the ranges reported by numerous other estimates as reviewed in ref. 62. The fraction of steel going to construction in individual countries is estimated to be 55% until they reach today's world average GDP/capita and then to decrease linearly to 25% to the UK's GDP/ capita.

The timber consumption on a country-by-country basis is obtained from the UN Food and Agriculture Organisation tables (FAOSTAT) for roundwood, coniferous and deciduous, subtracting exports and adding imports to production. The type of timber, deciduous or coniferous, is mostly determined by the local climate and does not affect use significantly. Roundwood is the type of timber which is mainly used in construction, though part of it is also used as fuel. The global fraction of harvested roundwood timber used in construction was estimated to be 10.9%, as described in the Supplementary Methods. For the range as plotted in Fig. 2, minimum and maximum values of 2.6% and 15.1% were used. The minimum bound of 2.6% corresponds to the share of roundwood in construction for countries which are not major producers of harvested wood products (first table within reference [63]). The maximum bound of 15.1% corresponds to the share of roundwood in construction for Japan (first table within reference [63]), a wealthy country with high timber consumption in construction.

The 2019 values for annual worldwide consumption of cement-based materials, steel and timber in construction, as calculated using the top-down method, are reported in Fig. 3a. This value is not an exact estimate, but a midpoint estimate within a likely range, given uncertainties around the average cement content used in cement-based materials, the proportion of steel used in construction, and the proportion of roundwood used in construction.

A master curve was generated from the historical data collated up to 2019. From this curve, future material consumption in 2050 was extrapolated based on projected GDP/capita in 2050. Plotting of the data and curve generation was carried out using the R software package.

The curves of per capita annual consumption against GDP/capita were plotted for individual countries, for cement (Supplementary Fig. 1) and steel (Supplementary Fig. 2). A master curve of cement, steel, and timber consumption per capita relative to the GDP per capita was generated, representative of the majority of national economies (Supplementary Fig. 3). This was derived from applying a LOESS filter from the R programme. Certain outlier countries were excluded from the dataset; a detailed explanation and list of excluded countries is provided in the Supplementary Methods and Supplementary Table 3.

The master curve (Supplementary Fig. 3) has higher uncertainty towards higher GDP/capita levels and was extrapolated linearly to 75,000\$ per capita from 50,000\$/capita. To extrapolate cement, steel and timber consumption on a country-by-country basis, the countries were assumed to keep a constant ratio of actual consumption to the master curve. This model assumes that the ratio of local consumption to the average worldwide consumption at a given GDP per capita is driven by local factors that will not vary significantly as the GDP per capita rises. The ratio is then applied to the master curve value at the extrapolated 2050 GDP.

## Bottom-up estimation approach

The demand for future construction materials from buildings was based on floor area projections by Deetman et al.[34]. The projections in the Deetman et al. dataset were carried out by a regression analysis of current building stock data, and projecting future material demand based on projections of several economic and demographic factors (see Marinova et al.[64] and Deetman et al.[34] for more details and a discussion of limitations). Despite some limitations, because this dataset is the only one so far which incorporates material inflows into future building stock projections that is disaggregated by region, it is a useful reference point.

Because the Deetman et al. projections do not consider infrastructure, estimates for construction material demand from infrastructure were carried out separately. For these two major categories of construction: buildings include use of materials in structural, non-structural and envelope functions (concrete, steel, mortar, timber, bricks), and infrastructure considers concrete, steel, mortar, asphalt, as described in the two following sections.

The bottom-up estimates for material consumption in buildings were based on projections aligned with global floor area growth from Deetman et al.[34]., which predicted an average 9.5 billion $m^2$ increase in the global floor area of buildings every year between 2025-2050. Material consumption was then estimated by multiplying representative values for global material intensity (kg/$m^2$ floor area) by the projected floor area increase. All calculations were carried out in Microsoft Excel 2021.

A database (Supplementary Table 4) was compiled to estimate the global material intensity (kg/$m^2$ floor area) in the current buildings stock as a weighted average of the different typologies (single unit, mid-rise and high-rise buildings), functions (commercial, industrial and residential) as well as primary material (concrete, steel or timber buildings). The underlying assumption, justified by the short-term future considered between 2025 and 2050, is that the share of each typology, function and primary material will stay constant (or change minimally) so that the material intensity averages are constant.

Material inputs for a 'median scenario' were calculated using average material intensity values from the literature dataset. Minimum and maximum values of material intensity were obtained from the range of literature values, to calculate a range of material inputs from buildings corresponding to a 'minimum scenario' and a 'maximum scenario' respectively.

Because the Deetman et al.[34]. projections do not consider infrastructure, demand for infrastructure materials were estimated pro-rata, using proportions for the global use of materials in buildings and infrastructure. Different calculation approaches were used for each material, based on whether they were used solely in buildings, or in buildings and infrastructure.

Asphalt is used almost solely in infrastructure, with negligible quantities used in buildings, so it was excluded from the bottom-up analysis. Bricks and timber were also assumed to be used solely in buildings, and not in infrastructure.

For cement-based materials, the situation is more complicated—whilst mortar is a major use of cement in buildings, it has very minor use in infrastructure, and hence was excluded from infrastructure. The

calculation was therefore done in two steps. Firstly, a global average was estimated for the split between concrete use in buildings and infrastructure (Supplementary Table 5); secondly, a global average was estimated for the split between cement use in concrete and other cement-based materials. Using these two splits, the total quantity of cement-based materials in construction was expressed as a factor of the quantity of concrete in buildings.

For steel, the global distribution of steel consumption by sector was estimated to be 0.358 Gt/year for buildings and 0.238 Gt/year for infrastructure in 2012 (Fig. 1 of[12]) – therefore, use of steel in buildings makes up an estimated 60% of the use of steel in all construction.

The proportion of material use in buildings, out of total use in construction, are given for each material in Supplementary Table 6. These values state the proportion used in buildings out of all construction (i.e., buildings and infrastructure); therefore, they differ from the values in Supplementary Table 1 which are the proportion in all construction out of total uses over all applications. Using these proportions, the demand for cement-based materials and steel from infrastructure were estimated pro-rata, factored on the demand from buildings.

Material demand from buildings only was calculated by multiplying the average material intensity values (Supplementary Table 4) by the projected annual floor area increase for the four most prominent materials by mass (i.e. cement-based materials, bricks, timber, steel). Values were calculated for 2019 and 2050: a likely range was obtained by using the minimum and maximum values of material intensity for each material (Supplementary Table 7).

Additional projected demand for cement-based materials and steel from infrastructure (explained above) was then added to obtain the total construction demand (Supplementary Table 8). Estimates for total material consumption of the cement-based materials, steel, and timber using the bottom-up approach were presented for 2019 and 2050 in Fig. 3a.

To obtain an estimate of how future demand for cement-based materials might vary between world regions, the approach above was extended to the 26 world regions defined in the IMAGE platform[11], using projections for annual floor area increase from 2019-2050 for each region[34]. As estimates for the split between cement-based materials use in buildings and infrastructure are not available for every region, the same global split (Supplementary Table 6) was applied to each region as an approximation. The estimated cumulative cement-based materials demand for 2019–2025 for each region is given in Supplementary Table 9; 2019 GDP/capita values for individual countries were the same used for the top-down analysis in Fig. 2, and used to generate a population-weighted average GDP/capita value for each of the 26 regions.

## Comparison of the potential of mitigation approaches

Figure 4 presents the cumulative effect of the $CO_2$ reducing interventions on the potential $CO_2$ emissions of global cement and steel production in 2050. Estimating mitigation potential to 2050 requires judgements about the materials or emissions reductions achievable for a given unit of structure or material. Given the subjective nature of such assumptions, it is important to describe the rationale and evidence base behind these judgments when estimating emission mitigation potential[65]. In the Supplementary Methods, detailed descriptions are given of how the mitigation potential for each strategy, and the upper and lower bounds, were estimated. The relative mitigation potential (i.e., % reduction) values are given in Supplementary Table 10. The absolute mitigation potential (i.e., Gt.$CO_2$/year reduction) values were were obtained by the multiplication of the various reduction factors for each strategy (Supplementary Table 10) to the extrapolated material demand using the master curve (Supplementary Fig. 3). Upper bounds and lower bounds for $CO_2$ emissions of global cement and steel production in 2050 were estimated,

assuming the most unfavourable penetration (or calculation method) and the most favourable, respectively.

### Reporting summary

Further information on research design is available in the Nature Portfolio Reporting Summary linked to this article.

## Data availability

The data used in this study are available in the Zenodo database under accession code https://doi.org/10.5281/zenodo.17604460. The data underpinning the projection assumptions used in this study are provided in the Supplementary Information.

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

## Acknowledgements

S.A.M. acknowledges funding from the National Science Foundation, grant CBET-2143981. The contents do not represent the official views or policies of the grantor or the State of California. F.Z. is supported by the Swiss National Science Foundation (SNSF) through an Ambizione fellowship (grant 208719). Thanks are given to Prof. Leon Black for valuable conversations about the relevance of the Environmental Kuznets Curve to cementitious materials.

## Author contributions

C.D. developed the approach and carried out analysis for the 'top-down approach' in Fig. 1 and carbon mitigation potential in Fig. 4; H.H. developed the approach and carried out analysis for the 'bottom-up approach' in Fig. 3; A.T.M.M. also developed the approach and carried out analysis for the 'bottom-up approach' in Fig. 3, and contributed to the estimates of materials quantities in the Introduction; S.A.M. developed the approach and carried out analysis for the estimates of material quantities, emissions sources and amounts, and ratios for use in Fig. 1; M.R. and W.S. provided feedback about the study design; K.S. developed the overall concept for the study and wrote the original draft; F.Z. contributed to estimates of carbon mitigation potential for concrete in Fig. 4. All co-authors contributed to the revision of the original draft.

## Competing interests

The Authors declare no competing interests.
