## [Transparent Peer Review file · Nature Communications]

Timely deployment of best-in-class technologies to enable development and decarbonise construction

Corresponding Author: Professor Karen Scrivener

Version 0:

Reviewer comments:

Reviewer #1

(Remarks to the Author)

This manuscript provides a thorough evaluation of available technologies for reducing emissions from the construction sector. It combines detailed bottom-up analyses with scenario modeling. The study is ambitious in scope and well-structured overall. However, several critical issues related to methodological justification, literature grounding, regional applicability, and interpretation of results need to be addressed for the manuscript to be ready for publication.

The authors estimate that emissions could decrease by up to 73% compared to a business-as-usual scenario. However, it is unclear how this level of reduction relates to the remaining global carbon budget under 1.5°C or 2°C scenarios. Is 73% sufficient? If not, what additional measures would be necessary? I believe that providing context around this issue would make the paper more relevant to climate policy discussions.

The manuscript has very limited engagement with existing studies that have explored similar questions about low-carbon construction pathways. Without a more robust literature review, the contribution of this paper is unclear. I suggest including a more thorough review of related work to clarify what is novel about this study and how it improves upon or differs from previous assessments.

The authors use GDP per capita as the primary explanatory variable to estimate future material use. However, this choice is not well justified. I suggest that the authors provide a theoretical justification for this approach.

While much of the previous literature uses material stock to estimate construction needs, this paper appears to focus on material flow. These two indicators reflect different system boundaries and temporal dynamics. I recommend that the authors justify the focus on material flows rather than stocks. Given that the authors have access to detailed bottom-up data, it is not clear why they chose to average bottom-up and top-down approaches.

The manuscript states that timber use is limited by the sustainably harvestable biomass. This is reasonable but somewhat simplistic. There are large unmanaged forest resources and better forest management could expand the available timber supply without necessarily converting new land.

The paper assumes a 25% global average reduction in material intensity through structural optimization. However, since engineering practices vary widely by region, this one-size-fits-all value may not reflect local potential or limitations. I recommend that the authors include region-specific assumptions or sensitivity ranges for this reduction factor.

The widespread use of LC3 is assumed to be feasible worldwide. However, this requires suitable clay deposits, which may not be evenly distributed. Please provide data supporting the geographic availability of calcined clays and the global feasibility of this assumption.

The manuscript claims that construction can absorb all recycled steel. This raises the issue of sectoral competition. If construction absorbs all available recycled or DRI-based steel, other sectors (e.g., automotive, manufacturing) may have to rely more on primary steel.

Although the paper covers many countries, it does not offer country-level insights or policy recommendations. This

undermines the value of the bottom-up modeling approach and limits its relevance to national-level decision-makers.

Reviewer #2

(Remarks to the Author)

I appreciate the opportunity to review this manuscript, which addresses the important and timely topic of decarbonisation in the building sector. The study presents a comprehensive assessment of mitigation strategies for reducing CO₂ emissions from cement-based materials and steel by 2050. The manuscript is generally well-structured and supported by a solid data framework. However, I have several comments and suggestions aimed at clarifying the scope of the study, strengthening the positioning of its contribution within the existing literature and improving the presentation of key assumptions and results. I hope these suggestions will help refine the manuscript.

While the manuscript addresses the important topic of decarbonisation in the building sector, I found the novelty of the work to be somewhat unclear. Specifically, it is not evident what new insights this study offers beyond incorporating infrastructure stocks into the analysis. A key prior study by Zhong et al. (2021) <https://doi.org/10.1038/s41467-021-26212-z> already quantified greenhouse gas emissions embodied in residential and commercial building materials through 2060 under both a baseline scenario and a High Efficiency (HE) scenario, incorporating a suite of material efficiency strategies. Notably, this study builds directly upon and continues the work by Deetman et al. (2020), which the authors also utilise in the current manuscript. Additionally, Pauliuk et al. (2021), although cited in the manuscript (line 129), is referenced only in the context of material stock estimation, without consideration of their findings related to potential CO₂ reduction. It would strengthen the manuscript if the authors clearly articulated how their study advances beyond these existing works and engaged more thoroughly with their results in the discussion.

The current title "Timely deployment of today's best in class construction technologies needed to allow countries to both develop and decarbonise" exceeds the journal's recommended limit of 15 words and should be shortened accordingly. In addition, the title is somewhat broad, particularly in its reference to "decarbonise," which may imply decarbonisation of all energy consumption. However, the study specifically addresses embodied emissions in construction and infrastructure materials, rather than broader energy-related emissions. I recommend that the authors revise the title to both meet the word limit and more accurately reflect the narrower focus of the study on material-related mitigation pathways. See the attached table!

While the methods section provides a detailed and generally clear description of the modelling framework and assumptions its current length is substantially above the typical limit recommended. In its current form, the methods read more like a technical report than a concise manuscript section. I recommend that the authors reduce the length by streamlining repetitive or overly detailed descriptions, potentially moving secondary methodological details to supplementary information. This would improve the overall readability of the manuscript without compromising transparency or reproducibility. This could also help address the issue of the manuscript exceeding the recommended reference count limit (currently over 70 references) and the figure/table count.

Line 66-67, you already mentioned that as a footnote.

Lines 80-88 Regarding your justification of the choice of strategies to be explored specifically line 88 the phrase "feasible, not yet widely deployed, technologies" could be clarified further. It should explicitly state that your selection criteria are not limited solely to technical feasibility, but also include for example economic and social feasibility. This distinction helps explain why certain widely studied and discussed strategies such as CCUS (already discussed lines 282–287) or the electrification of industrial processes (not only steel, but also in cement production where emission from energy consumption stands for about half of the impact) were excluded. Making this scope clear earlier in the manuscript, particularly in the introduction and possibly the title, would help readers better understand the rationale behind the selection of mitigation options.

Line 90 do you mean according to the categorisation by 7? Clarify.

Lines 207-222 & 836-859 Under the building structural design why did you not consider the decarbonisation potential of floor area reduction? See for example the recently published work by van Heerden et al. (2025) <https://doi.org/10.1038/s41560-025-01703-1>

There is an inconsistency in the use of British and American English throughout the manuscript. For example, the title uses "decarbonise" or "decarbonization" in line 55 (British spelling), while "decarbonization" is used in line 895 (American spelling).

Lines 252-260 support your text with references.

Line 297 Figure 5 what does the blue line indicate?

4.4.2 Given that the material intensity (MI) coefficient is a sensitive parameter and has been demonstrated in the literature to vary significantly over time particularly for cement, concrete and aggregates (CCA), as extensively documented by Heeren & Fishman et al. (2019) <https://doi.org/10.1038/s41597-019-0021-x> it is unclear why the authors did not consider potential

future changes in MI.

Line 363 under 4.2. How about the impact of insulation material production?

Line 460 CO_{2e} and CO_{2eq} in Line 348 for example. Choose one format!

Lines 850-859 Several claims throughout the paragraph are presented without supporting references.

Lines 850-854 This part i.e., the reuse potential, should be explored thoroughly. The disassembly and reuse are technically feasible and there are many examples of successful reuse projects documented in the literature. It is still more expensive than building with new elements, especially what you mentioned regarding testing to guarantee the element's performance. Nevertheless, the main reason why this circular economy strategy has negligible potential to reduce embodied carbon is the limited supply of reusable elements. Population growth, but worse than that, the increased consumption of building materials per person, limits the potential to close the loop. Here are a couple of references that could support this claim:

Al-Najjar, et al., 2025 <https://doi.org/10.1016/j.resconrec.2025.108229>

Zhu, et al., 2022 <https://doi.org/10.1016/j.enpol.2022.113222>

Lines 894-895 What I mentioned above makes me question this claim. Even though the embodied carbon reduction at project level is substantial, the limited supply of demolished concrete lowers the embodied carbon reduction potential. Is this what Dunant et al. whom you cited estimate? And why 5% reduction in line 896?

Lines 901-904 support the sentences with references.

Reviewer #3

(Remarks to the Author)

This is a very interesting paper, assessing a major challenge of our times: the apparent conflict between climate/environmental SDGs and development SDGs. I think the authors did a good job in analysing the problem and its potential solutions. Nevertheless I have some comments:

1. The UN has a panel since 2007, the International Resource Panel, that occupies itself with exactly this challenge. In their Global Resource Outlook of 2024, this is the main topic. Please have a look and at the least reference this publication (<https://www.unep.org/resources/Global-Resource-Outlook-2024>)
2. In the scenario assessments of this publication, the IMAGE + IMAGE-MAT models are actually used - an update of Deetman et al., which you use for your bottom-up estimates. I don't think the outcomes are that different, though, but anyway you can find the detailed assumptions in the annexes.
3. Nowadays there are some inventories of materials in infrastructure, please have a look for example at Engelenburg et al., the TRIPI database. It may not be directly usable, since it is a stocks inventory which is not (yet) connected to flows. It may be interesting for you to know that this is happening right now in the CIRCOMOD project, one of the EU-Horizon projects.
4. The present scenario assessments for construction often take a stocks-flows-service nexus approach: services (in this case, m² of useful floor area) are translated into stocks of buildings and the materials therein, which in turn are translated with a life span factor into flows of materials. The flows in this approach are derivatives of the stocks, and in assessments of future demand for materials this is an essential factor. This is perfectly illustrated by the case of China: they decided on building up their stocks and roughly 20 years later this was done, leading to a collapse of flows. Now, flows are related to the maintenance of these stocks and directly dependent on life spans (which in China are not that long, unfortunately). To relate this to GDP is not wrong, because such a decision can be made only when welfare has reached a certain height, but it is very indirect and ignores a crucial mechanism in society's metabolism. I would recommend at least to compare your forecasts with those based on stocks-flows-service nexus approaches.
5. In line with the previous comment, I miss one of the most powerful options to reduce flows while maintaining stocks and services: lengthening of the life span. I would suggest including this in your array of options. I also miss one option that has been highlighted in the work of the IRP: a sufficiency oriented measure, reducing the useful floor area in wealthier regions. See <https://www.resourcepanel.org/reports/technical-guidelines-resource-efficiency-andclimate-change-construction-sector>

Version 1:

Reviewer comments:

Reviewer #1

(Remarks to the Author)

I'd like to thank the authors for addressing all of my previous concerns. I believe the manuscript is now ready for publication. Congratulations to the authors!

Reviewer #2

(Remarks to the Author)

Thank you for the thorough revision. You addressed all my previous comments clearly and satisfactorily. I have no further suggestions.

Reviewer #3

(Remarks to the Author)

I am satisfied with the author's responses to my (reviewer#3) comments, with one exception: the suggestion to include lengthening of life span as a strategy. The authors argue that this is not relevant because of two reasons: (1) buildings are mostly demolished for other reasons than the end of their technical life span, and (2) the effects on flows of such lengthening will be visible only in the long term.

I agree that buildings are mostly demolished for other reasons. However, in my view that makes lengthening of life span an excellent strategy! It is a conscious decision to demolish, therefore, this decision could also be made differently - there is not even a need to adapt different building technologies.

The effects of such lengthening of life span will be visible straight away. For every building that is not demolished, a replacement is not necessary. The inflow of new materials can be reduced from day 1, while maintaining the building stock at current level. It's the recycling of demolition waste that has to wait, not the reduction of the inflow.

So I maintain my suggestion to include lengthening of lifespan in your array. If you decide against it, an alternative could be to dedicate a section in the discussion to this strategy.

Dear Reviewers,

Thank you for the various constructive suggestions for how to improve our manuscript, and the opportunity to revise it.

We have made our best efforts to positively respond to all the suggestions; particularly those around our methodological choices of using GDP as a key exploratory variable, and our focus on material flows rather than stocks. Given the broad scope of this study, there is a large quantity of potentially relevant literature that could be cited; in order to keep the manuscript concise and focussed, we have been selective in citing additional literature only when it is essential to address the methodological concerns identified; or, to highlight the novelty of our study's findings. We hope you find this balance reasonable and acceptable.

References within the manuscript and supplementary information use numerical format; references solely used within the 'response to reviewers' use Harvard format (to avoid confusion with overlapping numbering systems).

Kind regards,

Prof. Karen Scrivener and co-authors

REVIEWER COMMENTS

The authors would like to thank the reviewers for taking the time to provide thoughtful feedback on our work. We have provided point-by-point responses to the comments presented below.

Reviewer #1 (Remarks to the Author):

This manuscript provides a thorough evaluation of available technologies for reducing emissions from the construction sector. It combines detailed bottom-up analyses with scenario modeling. The study is ambitious in scope and well-structured overall. However, several critical issues related to methodological justification, literature grounding, regional applicability, and interpretation of results need to be addressed for the manuscript to be ready for publication.

Comment 1-1

The authors estimate that emissions could decrease by up to 73% compared to a business-as-usual scenario. However, it is unclear how this level of reduction relates to the remaining global carbon budget under 1.5°C or 2°C scenarios. Is 73% sufficient? If not, what additional measures would be necessary? I believe that providing context around this issue would make the paper more relevant to climate policy discussions.

R/ Thank you for this suggestion, we agree this is important useful global context. In the latest reporting round (AR6) (IPCC, 2023), IPCC suggests different reduction curves for different sector groups to keep global average warming below 1.5°C. For sector group 'transport, industry and buildings', the range of 45% to 87% reductions in emissions is required by 2050 (relative to emissions in 2015). In our study, the mid-point estimates for the adoption of selected mitigation strategies

projects total CO₂ emissions of ~2.8 Gt.CO₂/year – compared to the reference point of total emissions in 2019 (~6.1 Gt.CO₂/year), this represents a reduction of ~55%. This magnitude of reduction is hence compatible with the IPCC sector target reductions for the 1.5°C pathway scenario. Beyond this level of decarbonisation, to increase the likelihood of meeting the 1.5°C target and with an ultimate goal to achieve net zero, further reductions which are either more costly or more uncertain must be pursued.

We have added an additional statement in the Discussion section to put the findings of our study in this wider decarbonisation context:

L353-360: “Compared to a business-as-usual scenario in 2050, a 73% saving is achievable; however, taking into account the increase in demand between 2019 and 2050, the reduction in CO₂ emissions in comparison to the reference year of 2019 is ~55%. This value lies within the range of reductions (~45-87%) recommended for ‘transport, industry and buildings’ in the 1.5°C pathway from IPCC (i.e. a reduction of compared to 2015 levels)⁵². Whilst the ultimate goal remains Net Zero for construction materials, further decarbonisation will require more costly measures (e.g. CCUS) or more uncertain measures than depend on social factors (e.g. floor area reductions).

Comment 1-2

The manuscript has very limited engagement with existing studies that have explored similar questions about low-carbon construction pathways. Without a more robust literature review, the contribution of this paper is unclear. I suggest including a more thorough review of related work to clarify what is novel about this study and how it improves upon or differs from previous assessments.

R/ We have now added a conventional literature review section at the beginning of the manuscript, to more clearly describe what we perceive to be the knowledge gaps remaining in this field, which our study aims to address. We have also revised the Introduction to follow the more conventional style of an Introduction, and we moved Figure 1 and its descriptions into the Results section.

L36-75: “Decarbonisation of construction materials has received a notable amount of attention from both academic authors^{3,4} and industrial organisations^{5,6}, with a particular focus on cement & concrete and steel. Yet the specific question of what degree of decarbonisation is achievable at the global level, via readily implementable strategies, is less well-addressed. Zhong et al.⁷ modelled material decarbonisation achievable in the buildings sector (residential and commercial) up to 2060 for seven construction materials, using a range of seven material efficiency strategies; a reduction of ~62% was deemed achievable by 2050 (from a 2020 baseline of 3.17 Gt.CO₂/year to 1.21 Gt.CO₂/year in 2050). Pauliuk et al.⁸ focussed on residential buildings, and they studied the interactions of applying four materials efficiency strategies with different socio-economic pathways and background climate policies; for the reference policy and socio-economic pathway (i.e. SSP2), a reduction of ~49% was deemed achievable by 2050 (from 8.7 Gt.CO₂(eq.)/year in 2020 to 4.4 Gt.CO₂(eq.)/year in 2050). The larger absolute emissions values in this study are due to the inclusion of operational emissions in addition to embodied emissions. The Global Resource Outlook 2024⁹ took a holistic approach to decarbonisation pathways integrating material efficiency, energy policy and societal trends; whilst headline findings are clear, the broad scope makes it difficult to isolate the specific assumptions and mitigation contributions in construction materials production.

These previous global studies give an encouraging picture for decarbonisation, but they have two key limitations. First, studies used frameworks that account for material demand from buildings (i.e. RECC¹⁰ in Pauliuk et al.⁸, IMAGE¹¹ in Zhong et al.⁷), but not from infrastructure and other uses of the materials. Projecting material demands from infrastructure is more challenging, yet it is crucial for obtaining an accurate global picture. A significant global proportion of construction materials is already consumed in infrastructure (approximately 40% for steel¹², 33% for cement¹³), and large investments in new infrastructure are seen as essential to sustainable development in developing countries¹⁴. Second, some strategies evaluated are highly uncertain within the 25-year timeframe to 2050. For example, both Zhong et al.⁷ and Pauliuk et al.⁸ included reduction in per person floor area as an emissions reduction strategy; this is a potentially powerful material efficiency lever, but its implementation is dependent on highly uncertain social, policy and market factors¹⁵.

Two ongoing (yet unresolved) debates related to decarbonisation are the extent to which absolute decoupling is possible^{16,17}, and how pollution and consumption evolve with increasing national income (i.e. the disputed existence of the Environmental Kuznets curve^{18,19}). These debates are critical to questions around climate equity (i.e. how much should each country be allowed to emit?), and the synergies and trade-offs between decarbonisation and development (i.e. how should decarbonisation strategies differ for developing countries compared to developed countries?). Whilst the previous studies around material decarbonisation strategies do assess differences in stock and flow characteristics between different world regions and/or income brackets, they have not explicitly addressed how consumption trends relate to GDP per capita at the granularity of national level data.”

L95-98: “In this study, we aim to give a comprehensive answer to the question of what degree of decarbonisation in construction materials is possible using readily implementable strategies, already deployable within the materials and construction value chain. And, to contribute fresh insights to long-running debates around decoupling and development.”

Comment 1-3

The authors use GDP per capita as the primary explanatory variable to estimate future material use. However, this choice is not well justified. I suggest that the authors provide a theoretical justification for this approach.

R/ GDP does have some conceptual limitations, but nonetheless it has been widely used as an exploratory variable in material flow accounting (Krausmann et al., 2017) and in development economics, and particularly to explore questions around decoupling and consumption-income relationships (Haberl et al., 2020). We have summarised this in the updated literature review, included in the response to comment 1-2 above.

We have also added discussion around how our results (regarding trends with GDP) compare with previous findings and concepts in the material flow accounting and economic literature.

Added to Section 2.2 (i.e. top-down section):

L180-194: “The historical trends in Figure 2 show a relative decoupling of GDP and material use beyond a certain GDP per capita level, consistent with a general picture of relative decoupling conditions built up over numerous country-level studies¹⁷. Regardless of the point at which stocks saturate²⁸, the inflows of newly produced materials are the more critical issue in terms of decarbonisation. The ‘Environmental Kuznets Curve’ (EKC) postulates that the scale of environmental impacts follow an inverse U function with regards

to GDP per capita ²⁹. This concept has since been investigated with regards to materials consumption, with studies presenting evidence both in favour of ^{30,31} and against ³² the existence of a 'Materials Kuznets Curve'. The profile of the global master curves for steel and cement (Figure S3, SI) agrees well with the profile of the 'conventional EKC', in which increasing income leads to a decrease in impacts after a peak (but not a full decline) ¹⁹. This global trend, consistent with the EKC concept, also aligns with previous findings that in low- and medium-income countries, the gross capital formation is largely associated with urbanisation and building of infrastructure; whereas capital formation in medium-income and advanced economies takes the form of dematerialised investments and tooling ³³."

Comment 1-4

While much of the previous literature uses material stock to estimate construction needs, this paper appears to focus on material flow. These two indicators reflect different system boundaries and temporal dynamics. I recommend that the authors justify the focus on material flows rather than stocks. Given that the authors have access to detailed bottom-up data, it is not clear why they chose to average bottom-up and top-down approaches.

R/ The primary reason for focussing on material flows is that the main aim of our study is focussed on decarbonisation. Material flows directly describe the production quantities of construction materials, which hence partly determine the carbon emissions associated with their production. In contrast, whilst stocks are directly relevant to operational emissions, they are only indirectly relevant to embodied emissions of new materials production.

The secondary reason for focussing on material flows is due to some inherent limitations in our datasets. An aspect of novelty in our study is the inclusion of material demand for infrastructure, not just buildings. As described in the response to comment 1-1, this is important to obtain a more accurate projection of future construction materials demand. However, it also places some limitations on the type of analysis that is feasible. We build on the 'bottom-up' dataset developed by Deetman et al. for buildings, by applying a pro-rata value for additional material inflows for infrastructure. The close level of agreement between the 'top-down' and 'bottom-up' approaches indicates this approach is satisfactory for material flows. However, building a detailed global model for infrastructure stocks (as Deetman et al. did for buildings) is beyond the scope of this work. For these reasons, we have focussed our study on flows and not stocks.

We have added the following statements to clarify our approach:

Added to Section 1:

L104-106: "*We focus on material flows in this study, rather than stocks, as material flows are directly relevant to embodied carbon emissions of newly produced materials*".

Added to Section 2.3 (i.e. top-down section):

L180-184: "*The historical trends in Figure 2 show a relative decoupling of GDP and material use beyond a certain GDP per capita level, consistent with a general picture of relative decoupling conditions built up over numerous country-level studies ¹⁷. Regardless of the point at which stocks saturate ²⁸, the inflows of newly produced materials are the more critical issue in terms of decarbonisation.*"

Lastly, we realised we made an error in our description regarding the averaging of projections from the two approaches. In fact, the top-down projections were used for the subsequent carbon mitigation analysis. We have corrected this:

L246-247: “These calculations are based on the projected quantities established from the top-down approach above.”

Comment 1-5

The manuscript states that timber use is limited by the sustainably harvestable biomass. This is reasonable but somewhat simplistic. There are large unmanaged forest resources and better forest management could expand the available timber supply without necessarily converting new land.

R/ We believe there are two reasons why the sustainably harvestable biomass from existing forest area is quite limited. First, managed forests (defined as forests where human interventions and practices have been applied to perform production, ecological or social functions) represent 75% of the total forest area, and in many countries close to 100% (Grassi et al., 2018). So, the opportunity for converting unmanaged (i.e. primary) forest to managed forest is already highly restricted at the global scale. Second, primary forests are now recognised for being highly valuable and irreplaceable for their biodiversity (Gibson et al., 2011); their preservation as intact ecosystems is advocated as an urgent global priority (Watson et al., 2018). So, we deem the conversion of remaining unmanaged forests to managed forests to be undesirable for overall aims of sustainable development.

For the reasons outlined above, we do not think it is credible or desirable to advocate logging in primary forests as a global strategy. However, it may be that at in some regions, thinning practices and better forest management may make a small additional contribution to biomass that could be utilized.

Comment 1-6

The paper assumes a 25% global average reduction in material intensity through structural optimization. However, since engineering practices vary widely by region, this one-size-fits-all value may not reflect local potential or limitations. I recommend that the authors include region-specific assumptions or sensitivity ranges for this reduction factor.

R/ This value is not the result of local circumstances in structural design. The absolute, or characteristic value of the embodied carbon is strongly related to local circumstances. However, the relative value is linked to the process of design. Two key drivers of embodied carbon in design are:

- 1) The shape of the buildings. Denser dwellings are much more material efficient (Drewniok et al., 2023), with single storey dwellings having twice the embodied carbons of blocks of flats. The growing trends of urbanisation are therefore also trends towards material efficiency.
- 2) The choice of spans. This is a universal physical factor accounting for as much as 30% of superfluous carbon.

Furthermore, we refer to “structural optimization” within the constraints of compliance with existing codes. Whilst lower material intensities are possible with more exotic structural design approaches, we restrict the material intensity reductions to a level that is readily and widely achievable over the next 25 years. This is consistent with our updated definition of

“technically feasible and readily implementable” strategies (as given in our response to comment 2-5).

Comment 1-7

The widespread use of LC3 is assumed to be feasible worldwide. However, this requires suitable clay deposits, which may not be evenly distributed. Please provide data supporting the geographic availability of calcined clays and the global feasibility of this assumption.

R/ We have added evidence to support this statement in the main article:

Revised sentence in Section 2.4:

L316-319: *“However, new sources of substitute material are now becoming available using calcined clays and “Limestone Calcined Clay Cement” (LC3) technology⁴³; kaolinitic clays suitable for calcined clay production can be found within all world regions⁴⁴.”*

Comment 1-8

The manuscript claims that construction can absorb all recycled steel. This raises the issue of sectoral competition. If construction absorbs all available recycled or DRI-based steel, other sectors (e.g., automotive, manufacturing) may have to rely more on primary steel.

R/ It is well-known that the ‘low-quality steel’ used in construction has a higher tolerable content of tramp elements (i.e. impurities), and therefore can accept a higher proportion of recycled steel; whereas ‘high-quality steel’ used for cold-rolling flat plate (e.g. for use in the automotive sector) has a lower tolerable content of tramp elements, and therefore typically requires a lower proportion of recycled steel (and hence a higher proportion of primary steel) (Gao et al., 2025). It is possible to increase the proportion of ‘high quality steel’ recycled to make more high quality steel; however, under current conditions, it is realistic that the trend of construction being the main use for recycled steel continues over the next 25 years (Pauliuk et al., 2017).

Comment 1-9

Although the paper covers many countries, it does not offer country-level insights or policy recommendations. This undermines the value of the bottom-up modeling approach and limits its relevance to national-level decision-makers.

R/ We agree that country-level insights and policy recommendations would be useful, but national level recommendations are too complex and lengthy for a ‘communication’ type study on the global scale. Studies on material decarbonisation typically take a global, regional or country-level scope; it is unprecedented for a global study to then also give recommendations for individual countries. Although there will certainly be some regional nuances to deployment, most of the strategies investigated are widely applicable.

Reviewer #2 (Remarks to the Author):

I appreciate the opportunity to review this manuscript, which addresses the important and timely topic of decarbonisation in the building sector. The study presents a comprehensive assessment of mitigation strategies for reducing CO₂ emissions from cement-based materials and steel by 2050. The manuscript is generally well-structured and supported by a solid data framework. However, I have several comments and suggestions aimed at clarifying the scope of the study, strengthening the positioning of its contribution within the existing literature and improving the presentation of key assumptions and results. I hope these suggestions will help refine the manuscript.

Comment 2-1

While the manuscript addresses the important topic of decarbonisation in the building sector, I found the novelty of the work to be somewhat unclear. Specifically, it is not evident what new insights this study offers beyond incorporating infrastructure stocks into the analysis. A key prior study by Zhong et al. (2021) <https://doi.org/10.1038/s41467-021-26212-z> already quantified greenhouse gas emissions embodied in residential and commercial building materials through 2060 under both a baseline scenario and a High Efficiency (HE) scenario, incorporating a suite of material efficiency strategies. Notably, this study builds directly upon and continues the work by Deetman et al. (2020), which the authors also utilise in the current manuscript. Additionally, Pauliuk et al. (2021), although cited in the manuscript (line 129), is referenced only in the context of material stock estimation, without consideration of their findings related to potential CO₂ reduction. It would strengthen the manuscript if the authors clearly articulated how their study advances beyond these existing works and engaged more thoroughly with their results in the discussion.

R/ This comment is similar to comment 1-2 from reviewer 1. To address these points, we have added an evaluation of recent relevant literature in the Introduction (see response to comment 1-2) - this identifies the limitations of the previous studies in this area, and the novelty of this study in addressing these limitations.

We also make a contribution to the wider understanding about development economics, by showing a close agreement in the material trends with the conventional form of the 'Environmental Kuznets Curve' (see response to comment 1-3).

Lastly, as already highlighted in the Discussion and Abstract, a distinctive contribution of this study beyond the previous studies cited is our engagement with the political economy of decarbonisation. I.e. this effort creates a new foundation to make the case that decarbonisation and development are indeed compatible for developing countries (without radically changing development pathways).

Comment 2-2

The current title "Timely deployment of today's best in class construction technologies needed to allow countries to both develop and decarbonise" exceeds the journal's recommended limit of 15 words and should be shortened accordingly. In

addition, the title is somewhat broad, particularly in its reference to “decarbonise,” which may imply decarbonisation of all energy consumption. However, the study specifically addresses embodied emissions in construction and infrastructure materials, rather than broader energy-related emissions. I recommend that the authors revise the title to both meet the word limit and more accurately reflect the narrower focus of the study on material-related mitigation pathways. See the attached table!

R/ Based on these suggestions, we have revised the title as follows (now 13 words):

“Timely deployment of best-in-class technologies to enable development and decarbonise construction”

Comment 2-3

While the methods section provides a detailed and generally clear description of the modelling framework and assumptions its current length is substantially above the typical limit recommended. In its current form, the methods read more like a technical report than a concise manuscript section. I recommend that the authors reduce the length by streamlining repetitive or overly detailed descriptions, potentially moving secondary methodological details to supplementary information. This would improve the overall readability of the manuscript without compromising transparency or reproducibility. This could also help address the issue of the manuscript exceeding the recommended reference count limit (currently over 70 references) and the figure/table count.

R/ We have now streamlined the Methods section, so it follows the recommended <3000 word limit. In doing so, we have moved tables and detailed descriptions to Supplementary Information. The total number of references in the main manuscript is now <70.

Comment 2-4

Line 66-67, you already mentioned that as a footnote.

R/ We have now removed the definition of “cement-based materials” from the figure caption.

Comment 2-5

Lines 80-88 Regarding your justification of the choice of strategies to be explored specifically line 88 the phrase "feasible, not yet widely deployed, technologies" could be clarified further. It should explicitly state that your selection criteria are not limited solely to technical feasibility, but also include for example economic and social feasibility. This distinction helps explain why certain widely studied and discussed strategies such as CCUS (already discussed lines 282–287) or the electrification of industrial processes (not only steel, but also in cement production where emission from energy consumption stands for about half of the impact) were excluded. Making this scope clear earlier in the manuscript, particularly in the introduction and possibly the title, would help readers better understand the rationale behind the selection of mitigation options.

R/ Thank you for this suggestion – addressing this point helps us to clarify the scope of strategies we considered, as well as to better distinguish the novelty of our study from previous research in this topic.

Revision to Section 2.4 (mitigation approaches):

L241-245: *"In this section we look at the potential of technologies which are technically feasible and readily implementable today at a global scale, to lower the CO₂ emissions from cement-based materials and steel in 2050. This excludes previously investigated material efficiency strategies which we deemed to be either too unconventional for widespread adoption (e.g. novel structural systems) or to depend on external market, policy and social factors (e.g. reduction in per person floor area)."*

Comment 2-6

Line 90 do you mean according to the categorisation by 7? Clarify.

R/ We mean we have described the two approaches in line with the terminology used in the cited review on the topic. We have revised this sentence to clarify what was meant in the statement noted by the reviewer:

L98-100: *"To achieve this, we look at the potential evolution of the use of construction materials to 2050 from two complementary perspectives (using the top-down/bottom-up terminology by Lanau et al. ²³)."*

Comment 2-7

Lines 207-222 & 836-859 Under the building structural design why did you not consider the decarbonisation potential of floor area reduction? See for example the recently published work by van Heerden et al. (2025) <https://doi.org/10.1038/s41560-025-01703-1>

R/ We are aware of floor area reduction per person as a material efficiency strategy; we intentionally decided to exclude this strategy from consideration, as it does not fulfil our criteria for strategies that are *"technically feasible and readily implementable today at a global scale"* (as described in our response to comment 2-5). There are two main reasons behind our view. First, reduction of floor area per person as a *strategy* is not implemented within the materials and construction value chain – floor area person depends primarily on market and social factors, as described in (Lehner et al., 2024). Projecting future reductions in floor area would hence require underlying assumptions about the introduction of radical,

large-scale policy programmes. So, whilst reduction in floor area per person is possible and even desirable in developed regions, for the reasons above we do not consider it is feasible or viable within the next 25 years. Second, we do not consider it is a viable strategy on global scale. It is irrelevant to the large proportion of the global population which currently lives in overcrowded and inadequate housing conditions. For this reason, this strategy does not meet our criteria for being implementable at a global scale.

We expect that some readers may disagree with this view. Nonetheless, our restrictive scope around which strategies we consider to be “*technically feasible and readily implementable today at a global scale*” is a distinctive feature of our study compared to previous studies in this topic area – this is described in the updated introduction and literature review (as described in our response to comment 1-2).

Comment 2-8

There is an inconsistency in the use of British and American English throughout the manuscript. For example, the title uses “decarbonise” or “decarbonization” in line 55 (British spelling), while “decarbonization” is used in line 895 (American spelling).

R/ We have now revised for consistency in spelling.

Comment 2-9

Lines 252-260 support your text with references.

R/ We have updated this paragraph with references as advised:

*L321-330: “**Steel decarbonisation** (range assumed 42.5-90% of steel emissions): Steel can be electrically recycled in electric arc furnaces (EAF), and with grid emissions reducing, a likely practical limit of 100 kgCO₂/tonne can be achieved with the residual emissions linked to carbon injections, electrode consumption and lime use (detailed explanation in Section 4.4 in the Supplementary Information). This value is much lower than the current 1900 kg CO₂/tonne, which is the average (inc. Scope 1 and 2) emissions of steel in the world ⁴⁵. Recycled steel cannot be made in all grades, and it is frequently used for producing grades with the lowest strength requirements ⁴⁶. According to the IEA, 47-57% of steel will come from EAFs or DRI-EAFs in 2050 ⁵. Construction can use any type of steel and will be able to absorb all of the recycled steel produced ⁴⁷. Therefore, a practical factor of 65% abatement has been applied.”*

And further details added in the Supplementary Information, Section 4.4:

SI L449-460: “Direct CO₂ emissions (i.e. Scope 1) in the EAF process arises from oxidation of small amounts of carbon in the steel scrap mix, cathode consumption and the use of small amounts of quicklime – a reasonable upper value is 40 kg CO₂(eq.) / t.crude steel ⁹⁶. Electricity use for EAF steel production using scrap is approximately 500 kWh/t ⁹⁶. Scope 2 emissions are highly dependent on the emissions factor of electricity used. For the <1.5°C scenario, IPCC AR6 projects that global electricity production will reach net-zero sometime between 2044 and 2055 ⁹⁷. In a worst case scenario of keeping only within the <2°C scenario, electricity emissions are still projected to reduce from 2020 levels (461 g.CO₂/kWh ⁹⁸) by at least 75% in 2050 ⁹⁷, corresponding to a maximum of approximately 115 g.CO₂/kWh. Using this conservative value gives Scope 2 emissions of approximately 58 kg.CO₂ / t.crude steel. Summing the estimated Scope 1 and Scope 2 emissions described above gives a value of ~100 kg.CO₂ / t.crude steel (produced using scrap) in 2050, as used in the main article.

Comment 2-10

Line 297 Figure 5 what does the blue line indicate?

R/ The Figure 5 caption has been updated to explain the meaning of the black line:

L385-390: *“Country by country extrapolation of baseline projections under current trends from 2019 to 2050 of (A) cement consumption emissions per capita, and (C) steel consumption emissions per capita. The trends if all mitigation strategies considered in Figure 4 are applied are then presented for (B) cement and (D) steel. Green lines mark decreased emission, whereas red lines mark increased emissions. The black line on each plot represents the master curves for cement or steel, as shown in Figure S3 in the Supplementary Information.”*

Comment 2-11

4.4.2 Given that the material intensity (MI) coefficient is a sensitive parameter and has been demonstrated in the literature to vary significantly over time particularly for cement, concrete and aggregates (CCA), as extensively documented by Heeren & Fishman et al. (2019) <https://doi.org/10.1038/s41597-019-0021-x> it is unclear why the authors did not consider potential future changes in MI.

R/ Reduction in material intensity is implicitly incorporated across the mitigation strategies evaluated. This is primarily addressed within the ‘Building structural design’ strategy, as described in L273-290. To make this clear to readers familiar with the material intensity concept, we have updated this section:

L274-279: *“Many studies have shown that considerable reductions in material quantities (for the same floor area) are possible at the design stage for built systems – this lowers the material intensity coefficient (i.e. the mass of material per unit of floor area) of the structure, and hence lowers the embodied carbon of the structure. The choice of frame type, layout, and decking is commonly very suboptimal due to the wide range of possible choices and their comparatively low price sensitivity.”*

Comment 2-12

Line 363 under 4.2. How about the impact of insulation material production?

R/ Estimating the impacts of insulation materials is a greater challenge because this categorisation describes a function of material use and hence includes a diverse range of material types (e.g. mineral wool, natural wool, other, polymer foam), rather than one specific type of material. Perhaps as a result of this, we have noticed there is much less data around material intensity of insulation materials in buildings (compared to structural materials). For example, insulation materials are not included in the material demand projection developed by Deetman et al., which we have used for the floor projections in this study.

Therefore, we have added mineral wool to our analysis of materials in Figure 1, but have not added other types of insulation materials. We have also added plasterboard, and updated Figure 1 accordingly.

Additions in SI:

SI L195-206: “Plasterboard. The annual global production of gypsum plasterboard in 2019 was reported to be 10,836 Mm²/year, by area ²¹. To convert this into annual production volume, a representative thickness of 12.5 mm was assumed, which is a midpoint value of the common thicknesses sold ²². This gives an annual production volume of 135 Mm³. To convert annual production volume to annual production mass, a representative density of 640 kg/m³ was assumed - again, a midpoint value from common products ²². Multiplying annual production volume by density gives an annual production mass of 87 Mt/year. An embodied carbon value of 0.238 kg.CO₂(eq.) /kg was used, derived from the average of numerous EPDs reported in the ICE 4.0 Database ²³. Multiplying this value by the annual production mass gives the embodied carbon for annual plasterboard production of 21 Mt.CO₂(eq.)/year. The proportion of plasterboard used in construction was assumed to be 100%.

SI L207-219: “Mineral wool. Amongst types of insulation materials, mineral wool is the only material for which we deem there is sufficient data available to make a reasonable estimate of annual production quantities and emissions. The annual global production of mineral wool by mass in 2019 was reported to be 19 Mt/year ²⁴. To convert this into annual production volume, a representative density of 100 kg/m³ was used – this is a midpoint value for the common range of mineral wool densities (25 – 200 kg/m³) ²⁵. Dividing annual production mass by density gives an annual production volume of 190 Mm³/year. An embodied carbon value of 1.53 kg.CO₂(eq.) /kg was used, derived from the average of numerous EPDs reported in the ICE 4.0 Database ²³; this also matches well with the midpoint of the range established from EPD data in another study ²⁶. Multiplying this value by the annual production mass gives the embodied carbon for annual mineral wool production of 29 Mt.CO₂(eq.)/year. The proportion of mineral wool used in construction was assumed to be 100%.”

Comment 2-13

Line 460 CO₂e and CO₂eq in Line 348 for example. Choose one format!

R/ We have revised consistency with unit formats.

Comment 2-14

Lines 850-859 Several claims throughout the paragraph are presented without supporting references.

R/ We have updated this paragraph (now in the Supplementary Information) with references as advised, including the suggestions made in comment 2-15:

SI L393-417: “Reuse of concrete structural elements in new structures is another circular economy strategy which has the potential to displace production of primary concrete ⁸⁶. However, the feasibility of concrete reuse at present is mostly limited to pre-fabricated concrete elements ⁸⁷. As a result, the supply of concrete elements from end-of-life buildings is a major limiting factor for reuse; a study on Sweden, which has a mature building stock, found that even in recent years the inflows of pre-fabricated concrete elements were at least an order of magnitude greater than out flows ⁸⁸. In China, which has a less mature building stock, pre-fabricated buildings account for only 20% of new construction area ⁸⁹; even in the most restrictive scenario for new construction, authors estimated that only 3.8% of reinforced concrete could be reused in 2050.

Even within the limited available outflows, reuse of concrete elements is currently restricted by challenges around disassembly and validation; suitable testing procedures to guarantee structural performance are still in development⁹⁰. Whilst the number of concrete reuse case studies is growing⁹¹, the proportion of concrete reuse at the global concrete is negligible. Design-for-disassembly approaches can facilitate the reuse of concrete in future⁹²; however, even if such approaches were adopted universally immediately, the benefits of reuse could only be realised at the end of structures' service life, which should be ≥ 50 years.

Due to the previous reasons stated, current evidence suggests that the carbon mitigation potential of concrete reuse will be highly limited over 2025-2050. For the study on Sweden previously cited, the potential emissions savings from reuse of all eligible elements were calculated to have a maximum saving of 1% of lifecycle emissions⁸⁸. Thus, whilst concrete element reuse may be a valuable strategy to encourage over the longer term or in parts of the world where urban environments will rapidly be restructured, we do not include reuse within the group of strategies that we deem are technically feasible and readily implementable today at a global scale."

Comment 2-15

Lines 850-854 This part i.e., the reuse potential, should be explored thoroughly. The disassembly and reuse are technically feasible and there are many examples of successful reuse projects documented in the literature. It is still more expensive than building with new elements, especially what you mentioned regarding testing to guarantee the element's performance. Nevertheless, the main reason why this circular economy strategy has negligible potential to reduce embodied carbon is the limited supply of reusable elements. Population growth, but worse than that, the increased consumption of building materials per person, limits the potential to close the loop. Here are a couple of references that could support this claim:

Al-Najjar, et al., 2025 <https://doi.org/10.1016/j.resconrec.2025.108229>

Zhu, et al., 2022 <https://doi.org/10.1016/j.enpol.2022.113222>

R/ Thank you for these suggestions, we have used these to support our arguments in the revised passage included in our response to comment 2-14.

Comment 2-16

Lines 894-895 What I mentioned above makes me question this claim. Even though the embodied carbon reduction at project level is substantial, the limited supply of demolished concrete lowers the embodied carbon reduction potential. Is this what Dunant et al. whom you cited estimate? And why 5% reduction in line 896?

R/ We realise that the wording of this paragraph can be improved, to make it clearer which applications of concrete recycling are being described within the different sentences. We have provided more supporting references for our statements (as recommended in comment 2-17). We have also re-ordered this paragraph (now in the Supplementary Information), to first explain our rationale for the mitigation potential estimate used in Figure 3, and then to explain why we did not include other recycling routes as decarbonisation strategies.

SI L463-490: *"From end-of-life concrete, both the aggregate and cement paste can be recycled to produce secondary concrete, with a range of processing routes developed for both material outflow streams. In this study, we investigate the use of concrete fines as a replacement for raw meal in clinker production as a decarbonisation strategy (i.e. "Circular use of concrete fines"). Dunant et al.⁹⁹ estimated that the available amount of concrete fines*

by 2050 globally will amount to approximately 60% of clinker consumption. This means that the reuse of only a quarter of these fines in kilns, using the process currently deployed by Holcim, could easily abate 10% of the clinker carbon intensity

We do not consider that other routes for use of recycled concrete fit the criteria of being technically feasible and readily implementable today at a global scale; we describe our reasons for excluding these other routes in the paragraphs below.

Recycled cement paste can be subjected to an accelerated carbonation treatment, which makes it suitable for use as an SCM – the subsequent replacement of clinker using the recycled cement paste SCM hence offers mitigation potential ¹⁰⁰. However, this strategy has so far only been demonstrated at the laboratory-scale using synthetic hydrated cement paste ¹⁰⁰, so hence does not fulfil the criteria for being technically feasible and readily implementable today at a global scale.

Concrete at the end of life is commonly crushed and used as low quality aggregate. The use of recycled concrete aggregate is effective for reducing volumes of waste production and reducing raw material extraction. But due to the difficulty of completely removing hydrated cement paste from the surface of recycled concrete aggregate, more slightly water and cement is required for use of recycled concrete aggregates in concrete (for equivalent concrete strength class using natural aggregates); this results in a small increase in embodied carbon ¹⁰¹. Whilst recycled concrete aggregate can offer modest carbon savings over natural aggregate in some specific situations, this is highly dependent on transport distances ¹⁰¹. As a result, greater use of recycled concrete aggregate is not expected to lead to any meaningful carbon mitigation at the global scale, and so we have not considered it as a decarbonisation strategy.”

Comment 2-17

Lines 901-904 support the sentences with references.

R/ We have updated this paragraph with references – see response to comment 2-16 for revised text.

Reviewer #3 (Remarks to the Author):

This is a very interesting paper, assessing a major challenge of our times: the apparent conflict between climate/environmental SDGs and development SDGs. I think the authors did a good job in analysing the problem and its potential solutions. Nevertheless I have some comments:

Comment 3-1

The UN has a panel since 2007, the International Resource Panel, that occupies itself with exactly this challenge. In their Global Resource Outlook of 2024, this is the main topic. Please have a look and at the least reference this publication (<https://www.unep.org/resources/Global-Resource-Outlook-2024>)

R/ Thank you for suggesting this source, we agree it is a highly relevant study to consider. We have now compared our findings to it, for material production projections and mitigation potential:

Comparison of baseline scenario projections:

L215-218: *“The scale of increase (~43%) in consumption of cement-based materials to 2050 is similar to that projected by the Global Cement and Concrete Association ⁶, and not far from the ~60% increase of construction material inflows into construction in the Global Resources Outlook 2024 ⁹.*

Comparison of mitigation potential of readily available technological strategies:

L360-368: *“In a global evaluation considering all sources of GHG emissions, Global Resource Outlook 2024 projected that by 2060 (relative to a reference year of 2020), the increase in emissions due to increasing affluence and population growth (+62.8 Gt.CO₂(eq.)/year) would largely be cancelled out by reductions in emissions due to current technology trends (-51.0 Gt.CO₂(eq.)/year) ⁹. As a consequence, net reductions (with regard to the 2020 reference year) were only deemed achievable through policy and societal shifts. In contrast, our study indicates that within the sector of construction materials a much higher degree of relative decoupling is achievable within current feasible technologies, beyond which strategies relying on major policy shifts and societal changes are needed.”*

Comment 3-2

In the scenario assessments of this publication, the IMAGE + IMAGE-MAT models are actually used - an update of Deetman et al., which you use for your bottom-up estimates. I don't think the outcomes are that different, though, but anyway you can find the detailed assumptions in the annexes.

R/ Thank you for bringing the update of the materials projections in IMAGE to our attention. Due to the complexity of the approach here we have decided to use the original database for this study. We will follow this development with interest.

Comment 3-3

Nowadays there are some inventories of materials in infrastructure, please have a look for example at Engelenburg et al., the TRIPI database. It may not be directly usable, since it is a stocks inventory which is not (yet) connected to flows. It may be

interesting for you to know that this is happening right now in the CIRCOMOD project, one of the EU-Horizon projects.

R/ TRIPI database does include projections for transport infrastructure, but it has two key limitations for incorporation in this study. First, as noted by the reviewer, it is only stocks (for now). Second, it includes transport infrastructure, but not other kinds of infrastructure (e.g. ports, dams, other energy infrastructure). So, in its current form it only gives a part of the overall picture of global flows of construction materials going to infrastructure.

Nonetheless, we have acknowledged the existence of this dataset, and the potential for future developments to improve the accuracy of projecting construction material demand for infrastructure. We look forward to seeing the outcomes of the CIRCOMOD project.

Addition to Supplementary Information, Section 3:

SI L256-260: *“Whilst there have been some data published around global material stocks in infrastructure ⁶³, there are not yet any comprehensive datasets for global flows of construction materials into infrastructure. Future developments in this field could be used to update and improve the approach used in this study.”*

Comment 3-4

The present scenario assessments for construction often take a stocks-flows-service nexus approach: services (in this case, m² of useful floor area) are translated into stocks of buildings and the materials therein, which in turn are translated with a life span factor into flows of materials. The flows in this approach are derivatives of the stocks, and in assessments of future demand for materials this is an essential factor. This is perfectly illustrated by the case of China: they decided on building up their stocks and roughly 20 years later this was done, leading to a collapse of flows. Now, flows are related to the maintenance of these stocks and directly dependent on life spans (which in China are not that long, unfortunately). To relate this to GDP is not wrong, because such a decision can be made only when welfare has reached a certain height, but it is very indirect and ignores a crucial mechanism in society's metabolism. I would recommend at least to compare your forecasts with those based on stocks-flows-service nexus approaches.

R/ Thank you for the suggestion. The justification for focussing on material flows was also raised in comment 1-4. In addition to the point in our response to that comment about the current limitations in accurate modelling of infrastructure stocks, the same argument also applies for service units of infrastructure. Whilst m² of floor area works well as a service unit for buildings, we are not aware of any similar service unit for infrastructure, due to the vastly different functions of different infrastructure types (e.g. bridges, energy, ports). We hope that in future other research studies will help flow-stock-service modelling for infrastructure ‘catch up’ with buildings, but for now this is beyond the scope of our study.

We have added a comparison with the projections from the Global Resources Outlook 2024:

L215-218: *“The scale of increase (~43%) in consumption of cement-based materials to 2050 is similar to that projected by the Global Cement and Concrete Association ⁶, and not far from the ~60% increase of construction material inflows into construction in the Global Resources Outlook 2024 ⁹.”*

Comment 3-5

In line with the previous comment, I miss one of the most powerful options to reduce flows while maintaining stocks and services: lengthening of the life span. I would suggest including this in your array of options. I also miss one option that has been highlighted in the work of the IRP: a sufficiency oriented measure, reducing the useful floor area in wealthier regions.

See <https://www.resourcepanel.org/reports/technical-guidelines-resource-efficiency-andclimate-change-construction-sector>

R/ Technical/physical obsolescence (i.e. the point of breakdown beyond viable repair) is rarely why structures reach end-of-life, as there are also numerous other reasons for obsolescence i.e. functional, economical, legal and desirability (Ashby, 2013). Because of this, it is unfortunately a widespread reality that technically sound buildings are demolished well before their technical end-of-life. Because the building lifespan depends on the numerous factors listed above, lengthening of the lifespan does not fulfil our criteria for strategies that are “technically feasible and readily implementable today at a global scale” (as described in our response to comment 2-5). Therefore, we excluded lengthening of lifespan from the scope of our study (for similar reasons to reduction in floor area per person, as described in our response to comment 2-7).

Notwithstanding our methodological reasons for excluding lifespan extension as a strategy, material use applications that are longer-lived (e.g., buildings) tend to not offer as much of an immediate benefit to lengthening life span as shorter-lived applications (Miller, 2020). In cases where immediate reductions in material production can be achieved through prolonged use and reduced replacement (e.g., using a cell phone for 3 years instead of 2), emissions reductions can potentially be seen on an appreciable scale. For the built environment, the long time horizon for replacement indicates emissions savings may not accrue for decades after initial installation. This reduces the relevance to the short-term focus of this study up to 2050.

References (included in 'response to reviewers' only)

- ASHBY, M. F. 2013. Chapter 4 - End of first life: A problem or a resource? *In: ASHBY, M. F. (ed.) Materials and the Environment (Second Edition)*. Boston: Butterworth-Heinemann.
- DREWNIOK, M. P., DUNANT, C. F., ALLWOOD, J. M., IBELL, T. & HAWKINS, W. 2023. Modelling the embodied carbon cost of UK domestic building construction: Today to 2050. *Ecological Economics*, 205, 107725.
- GAO, H., LIU, J. & DAIGO, I. 2025. Methodology development for estimating the impact of restriction factors to promote national steel recycling. *Resources, Conservation and Recycling*, 215, 108052.
- GIBSON, L., LEE, T. M., KOH, L. P., BROOK, B. W., GARDNER, T. A., BARLOW, J., PERES, C. A., BRADSHAW, C. J. A., LAURANCE, W. F., LOVEJOY, T. E. & SODHI, N. S. 2011. Primary forests are irreplaceable for sustaining tropical biodiversity. *Nature*, 478, 378-381.
- GRASSI, G., HOUSE, J., KURZ, W. A., CESCATTI, A., HOUGHTON, R. A., PETERS, G. P., SANZ, M. J., VIÑAS, R. A., ALKAMA, R., ARNETH, A., BONDEAU, A., DENTENER, F., FADER, M., FEDERICI, S., FRIEDLINGSTEIN, P., JAIN, A. K., KATO, E., KOVEN, C. D., LEE, D., NABEL, J. E. M. S., NASSIKAS, A. A., PERUGINI, L., ROSSI, S., SITCH, S., VIOVY, N., WILTSHIRE, A. & ZAEHLE, S. 2018. Reconciling global-model estimates and country reporting of anthropogenic forest CO₂ sinks. *Nature Climate Change*, 8, 914-920.
- HABERL, H., WIEDENHOFER, D., VIRÁG, D., KALT, G., PLANK, B., BROCKWAY, P., FISHMAN, T., HAUSKNOST, D., KRAUSMANN, F., LEON-GRUCHALSKI, B., MAYER, A., PICHLER, M., SCHAFFARTZIK, A., SOUSA, T., STREECK, J. & CREUTZIG, F. 2020. A systematic review of the evidence on decoupling of GDP, resource use and GHG emissions, part II: synthesizing the insights. *Environmental Research Letters*, 15, 065003.
- IPCC 2023. Climate Change 2023 Synthesis Report. *In: CORE WRITING TEAM, LEE, H. & ROMERO, J. (eds.) Contribution of Working Groups I, II and III to the Sixth Assessment Report of the Intergovernmental Panel on Climate Change*. Geneva, Switzerland: IPCC.
- KRAUSMANN, F., SCHANDL, H., EISENMENGER, N., GILJUM, S. & JACKSON, T. 2017. Material Flow Accounting: Measuring Global Material Use for Sustainable Development. *Annual Review of Environment and Resources*, 42, 647-675.
- LEHNER, M., RICHTER, J. L., KREININ, H., MAMUT, P., VADOVICS, E., HENMAN, J., MONT, O. & FUCHS, D. 2024. Living smaller: acceptance, effects and structural factors in the EU. *Buildings and Cities*.
- MILLER, S. A. 2020. The role of cement service-life on the efficient use of resources. *Environmental Research Letters*, 15, 024004.
- PAULIUK, S., KONDO, Y., NAKAMURA, S. & NAKAJIMA, K. 2017. Regional distribution and losses of end-of-life steel throughout multiple product life cycles—Insights from the global multiregional MaTrace model. *Resources, Conservation and Recycling*, 116, 84-93.
- WATSON, J. E. M., EVANS, T., VENTER, O., WILLIAMS, B., TULLOCH, A., STEWART, C., THOMPSON, I., RAY, J. C., MURRAY, K., SALAZAR, A., MCALPINE, C., POTAPOV, P., WALSTON, J., ROBINSON, J. G., PAINTER, M., WILKIE, D., FILARDI, C., LAURANCE, W. F., HOUGHTON, R. A.,

MAXWELL, S., GRANTHAM, H., SAMPER, C., WANG, S., LAESTADIUS, L., RUNTING, R. K., SILVA-CHÁVEZ, G. A., ERVIN, J. & LINDENMAYER, D. 2018. The exceptional value of intact forest ecosystems. *Nature Ecology & Evolution*, 2, 599-610.

Reviewer 3 comment:

I am satisfied with the author's responses to my (reviewer#3) comments, with one exception: the suggestion to include lengthening of life span as a strategy. The authors argue that this is not relevant because of two reasons: (1) buildings are mostly demolished for other reasons than the end of their technical life span, and (2) the effects on flows of such lengthening will be visible only in the long term.

I agree that buildings are mostly demolished for other reasons. However, in my view that makes lengthening of life span an excellent strategy! It is a conscious decision to demolish, therefore, this decision could also be made differently - there is not even a need to adapt different building technologies.

The effects of such lengthening of life span will be visible straight away. For every building that is not demolished, a replacement is not necessary. The inflow of new materials can be reduced from day 1, while maintaining the building stock at current level. It's the recycling of demolition waste that has to wait, not the reduction of the inflow.

So I maintain my suggestion to include lengthening of lifespan in your array. If you decide against it, an alternative could be to dedicate a section in the discussion to this strategy.

Authors' response:

We agree about the merits of lengthening life span. We excluded this from the array of strategies suggested, as it is primarily a non-technical lever, that depends more on economically and socially driven choices. We applied the same criteria to habitable floor area; it is indeed a decarbonisation lever, but it is primarily not a technical lever within the material and construction supply chain, and is also mostly driven by economic and social factors.

In response, we have added an additional statement to the discussion, to explain that this lever has potential (with more uncertainty) above the strategies explicitly investigated in this study:

L348-351: "For example, extending the lifespan of buildings to reach closer to their technical lifespan; this will require tackling social and economic incentives for premature demolition, especially for high-income countries with broadly stable building stocks."